# Early-stage bifurcation of crystallization in a sphere

Chrameh Fru Mbah [1,5], Junwei Wang [2,5], Silvan Englisch [3,5], Praveen Bommineni [1,4], Nydia Roxana Varela-Rosales [1], Erdmann Spiecker [3], Nicolas Vogel [2] ✉ & Michael Engel [1] ✉

Bifurcations in kinetic pathways decide the evolution of a system. An example is crystallization, in which the thermodynamically stable polymorph may not form due to kinetic hindrance. Here, we use confined self-assembly to investigate the interplay of thermodynamics and kinetics in the crystallization pathways of finite clusters. We report the observation of decahedral clusters from colloidal particles in emulsion droplets and show that these decahedral clusters can be thermodynamically stable, just like icosahedral clusters. Our hard sphere simulations reveal how the development of the early nucleus shape passes through a bifurcation that decides the cluster symmetry. A geometric argument explains why decahedral clusters are kinetically hindered and why icosahedral clusters can be dominant even if they are not in the thermodynamic ground state.

Many crystalline materials exist in polymorphs with structure-dependent properties. Certain pathways are required to form a desired polymorph, to transition from one polymorph to another, and to shape crystals[1,2]. It is generally not well understood which factors affect the formation of specific polymorphs and whether thermodynamics or kinetics dominate in the process. Thermodynamics is important because it predicts equilibrium crystal structures. Yet not all thermodynamically stable crystal structures are experimentally observed. Kinetics must also be considered to determine whether there is a reliable and robust pathway to form a thermodynamically stable crystal structure.

More complications arise when the crystal is small. For example, the equilibrium structure of bulk gold is the face-centered cubic (fcc) lattice, which in nanocrystals typically results in a truncated octahedron Wulff shape. Such truncated octahedra are highly facetted and expose different crystal planes at their surface. Since these crystal planes can have different surface free energies, gold nanocrystals can show a tendency to expose more facets with high coordination as a strategy to reduce free energy. These surface free energy considerations explain the discovery of nanoclusters with multiply-twinned icosahedral or decahedral configurations[3–5]. It is now possible to reliably and reproducibly crystallize icosahedral or decahedral nanoparticles by seeding or surfactant functionalization in wet-chemical synthesis[6,7]. Still, the competition between icosahedral and decahedral symmetry in nanoparticles remains poorly understood[8,9].

The framework of magic number clusters describes the geometry of finite clusters. It is found that free energy fluctuates as a function of the number of particles. Magic number clusters correspond to free energy minima and appear whenever the number of building blocks suits the formation of closed shells[10,11], which maximizes the coordination number at the surface. Because icosahedral and decahedral nanoparticles endorse different sets of magic numbers, the preferred crystal structure is often predicted by considering thermodynamics at different numbers of particles[12]. From a kinetic point of view, however, as the system starts from an isotropic fluid and ends in two anisotropic crystals, it must undergo a bifurcation at some point during the crystallization process. At this bifurcation, the growing crystal decides whether to evolve into icosahedral symmetry or decahedral symmetry.

[1]Institute for Multiscale Simulation, IZNF, Friedrich-Alexander-Universität Erlangen-Nürnberg, 91058 Erlangen, Germany. [2]Institute of Particle Technology, Friedrich-Alexander-Universität Erlangen-Nürnberg, 91058 Erlangen, Germany. [3]Institute of Micro- and Nanostructure Research and Center for Nanoanalysis and Electron Microscopy, IZNF, Friedrich-Alexander-Universität Erlangen-Nürnberg, 91058 Erlangen, Germany. [4]Department of Chemical Engineering, National Institute of Technology Warangal, Warangal, Telangana 506004, India. [5]These authors contributed equally: Chrameh Fru Mbah, Junwei Wang, Silvan Englisch. ✉e-mail: nicolas.vogel@fau.de; michael.engel@fau.de

Here, we study the role of thermodynamics and kinetics for bifurcation in colloidal crystallization. Colloidal particles are a classical model system for condensed matter to study phase transitions and structure formation[13]. The large size of colloidal particles allows both the direct observation of crystal structures via optical microscopy[14] and to follow the crystallization pathways in situ via structural coloration[15]. Colloidal particles are also appealing because the complex atomic interactions can be reduced to hard sphere-like repulsion, which is well suited for computer simulation. And while the phase behavior of hard spheres is dominated by entropy[16], the hard sphere model retains complex nucleation pathways, including polymorphism[17]. The analogy between atoms and colloids can be transferred to finite clusters by confining colloidal particles within an emulsion droplet[18–25]. Similar to nanoparticles, colloidal clusters favor icosahedral symmetry over the bulk fcc structure[18,19] and feature magic number configurations with closed-shell structures as free energy minima[19,20]. In this contribution, we report the experimental observation of decahedral colloidal clusters. We show that thermodynamics is not sufficient to explain the outcomes of experiments. Instead, kinetic factors are important. Finally, we introduce a simple geometric argument that explains the dominance of icosahedral colloidal clusters found in experiment by considering the shape of the growing nucleus.

## Results

### Discovery of decahedral clusters

We fabricate colloidal clusters by drying droplets of an aqueous polystyrene (PS) particle dispersion in a continuous oil phase[19]. Slow drying provides sufficient equilibration time for the system to undergo a phase transition from the isotropic fluid phase to two distinct classes of solid with fivefold symmetries (Fig. 1a). As in previous works[18,19], we find many icosahedral clusters. Icosahedral clusters are the dominant colloidal cluster polymorph. They consist of 20 slightly deformed, quasi-tetrahedral fcc grains that create 20 hexagonal (111) surface patches on the cluster surface (Fig. 1b). A careful examination also finds another cluster polymorph accompanying the icosahedral clusters, decahedral clusters. Decahedral clusters have five hexagonal (111) patches on their surface surrounding a fivefold symmetry axis and present five square-shaped (100) patches perpendicular to the fivefold axis (Fig. 1c, Supplementary Figs. S1 and S2). While colloidal clusters of

both symmetries are present across all cluster sizes, the occurrence of decahedral clusters is generally much lower with $(2.0 \pm 1.4)\%$ decahedral cluster for clusters with approximately 10000 particles and $(16.2 \pm 4.5)\%$ decahedral clusters for clusters with approximately 35,000 particles (Fig. 1d). This may explain why their existence remained elusive in earlier investigations of colloidal systems.

Decahedral clusters can be understood as packings of equal-sized spheres in five fcc grains with slight deformations twinned around a single fivefold axis (Supplementary Figs. S1 and S2). This is a key structural difference compared to icosahedral clusters, which have six fivefold axes. We count the number of fivefold axes to identify cluster symmetry by visual inspections of its surface. The characteristic surface features, especially the anti-Mackay type surface twinning in some bimetallic nanoparticles[26], can be modeled accurately by applying spherical truncation to the sphere packing model (Fig. 1c, Supplementary Figs. S3 and S4)[4,18,19,27]. We resolve the interior of a decahedral colloidal cluster by transmission X-ray imaging[28] (Fig. 1e). Five grains appear twinned around a central fivefold axis. Tomographic reconstruction supplies even more insights. We resolve 2286 primary particles forming a decahedral cluster with a central pentagonal bipyramid (Supplementary Fig. S5, Supplementary Movies 1–3), unambiguously confirming the geometric model in Fig. 1c.

After having established the existence of decahedral clusters in experiment, we now want to understand their lower frequency of occurrence compared to icosahedral clusters. For this purpose, we switch our focus to the hard sphere model system and study hard spheres in hard spherical confinement via computer simulations. We use particle simulations to explore the cluster geometry and free energy calculations to reveal their thermodynamic properties.

### Relative thermodynamic stability of icosahedral and decahedral clusters

To understand the relative thermodynamic stability of icosahedral and decahedral clusters, we calculate free energies with high precision using a simulation framework with harmonic springs and Monte Carlo swaps of spring developed in our previous work[19] (see also "Methods"). Calculations are performed for ideal icosahedral model clusters (blue in Fig. 2a, including both Mackay and anti-Mackay type, Supplementary Fig. S6) and ideal decahedral model clusters (yellow). As expected,

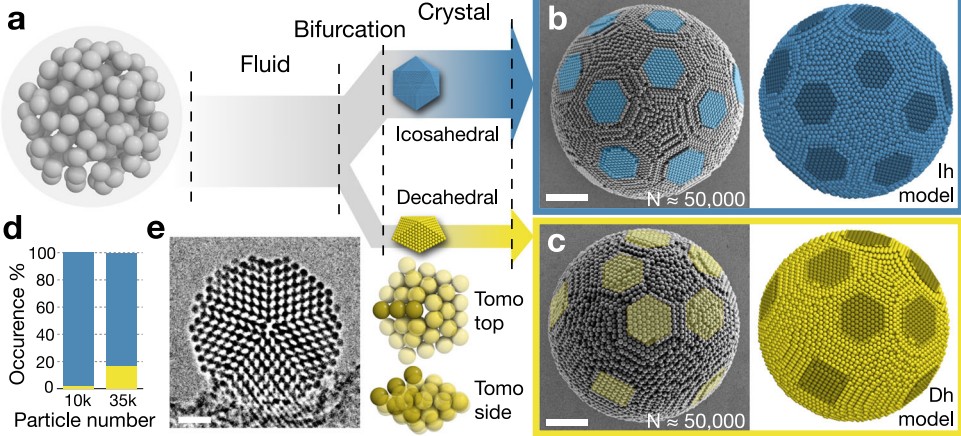

**Fig. 1 | Structure characterization of icosahedral and decahedral colloidal cluster polymorphs observed in experiment. a** Colloidal particles in spherical confinement crystallize into clusters with icosahedral and decahedral symmetry. **b** Icosahedral clusters (Ih) are characterized by the presence of multiple fivefold symmetry axes. **c** Decahedral clusters (Dh) are characterized by only one fivefold symmetry axis at their surface. Scanning electron microscopy observations (left in **b, c**) are compared to ideal structure models (right in **b, c**). **d** The occurrence of decahedral clusters is significantly lower than the occurrence of icosahedral

clusters. A total number of 100 (68) clusters, among them 2 (11) decahedral clusters (yellow color), were analyzed for clusters with particle number 10k (35k).
**e** Transmission X-ray image along the fivefold axis and segmented central region of tomographic reconstruction (Tomo, Supplementary Fig. S5) confirm the presence of decahedral symmetry by revealing particles columns (dark contrast) and their fcc arrangement with slight deformation in five twinned grains. Due to positive Zernike phase contrast, particle columns appear dark while the empty interstices appear bright. All scalebars, 2 μm.

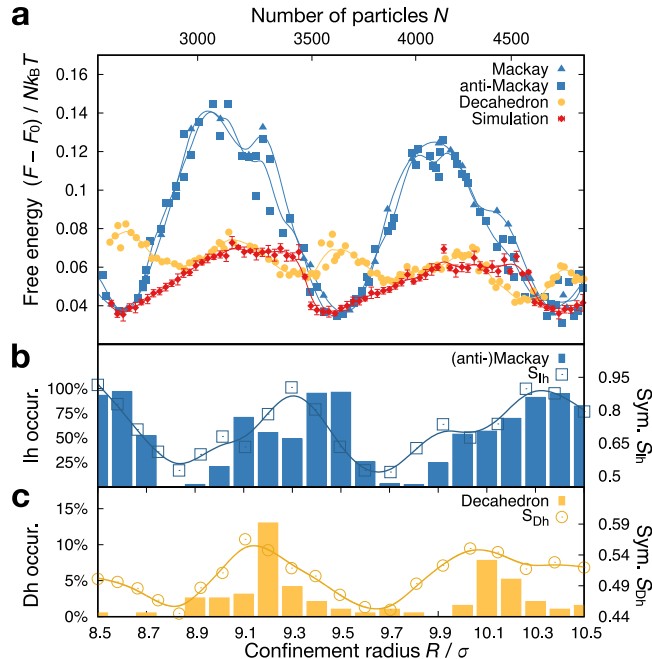

**Fig. 2 | Thermodynamics of icosahedral and decahedral hard sphere clusters from numerical calculations. a** Free energies calculated for icosahedral model clusters (blue, Mackay and anti-Mackay types), decahedral model clusters (yellow), and simulated clusters (red). Data points are free energy values ((anti-)Mackay and decahedral clusters) and averaged over five free energy values (simulated clusters; with standard errors). Lines are guides to the eyes. Free energies fluctuate with the number of particles and exhibit characteristic minima. The number of particles N is controlled by confinement radius in units of the sphere diameter $\sigma$. In all free energy calculations and simulations, packing fraction is fixed at 52%.

**b**, **c** Occurrence statistics of icosahedral (Ih, **b**) and decahedral (Dh, **c**) clusters for 21 confinement radii (solid bars). 192 simulations were performed at each confinement radius. The occurrence of Ih and Dh does not add up to 100% because defected icosahedral clusters with weakly broken icosahedral symmetry[15] are not classified as icosahedral clusters. Point group symmetry is quantified at the end of each simulation run and averaged over all simulations at a given system size independent of cluster symmetry. Open symbols show point group order parameters (see "Methods"). Lines are guides to the eye.

magic number effects[19] cause significant fluctuations of free energy of clusters of both symmetries. Furthermore, the free energy of decahedral clusters is found to fluctuate with a higher frequency and a lower amplitude than the free energy of icosahedral clusters. The higher frequency of fluctuation for decahedral clusters indicates the presence of more magic numbers, which is expected from the lower symmetry of decahedral clusters. Importantly, our results demonstrate that not all ground states of confined hard spheres are icosahedral clusters. For some numbers of particles, for example, near the confinement radii $9.3\sigma$ and $10.2\sigma$, free energy calculation predicts a thermodynamic preference for decahedral clusters.

We also perform free energy calculations for hard sphere clusters that were crystallized from the melt in event-driven molecular dynamics simulations at constant packing fraction. These clusters, termed 'simulated clusters,' constitute our closest computational analogs to the experimental colloidal clusters. The free energy curve of the simulated clusters (red in Fig. 2a) traces the free energy minima of icosahedral model clusters closely but does not entirely trace the free energy minima of decahedral model clusters. Apparently, crystallization in hard sphere simulation consistently fails to reach ground states with decahedral symmetry. This is the case even at size ranges where decahedral clusters are thermodynamically stable.

Our findings indicate that it is difficult to find decahedral clusters in experiments with colloids. But how difficult is it? We evaluate the occurrence of cluster polymorphs statistically and systematically in simulation. We perform in total 4032 crystallization simulations across system sizes from 2000 to 5000 hard spheres. The symmetries of the obtained cluster are systematically classified with bond-orientational order parameters (Supplementary Figs. S7 and S8) in conjunction with manual inspection. We perform the classification for all clusters with decahedral symmetry and high icosahedral symmetry, omitting in our analysis defected clusters with weakly broken icosahedral symmetry. These defected icosahedral clusters dominate in the off-magic number regions, where closed-shell clusters cannot be formed due to the accumulation of defects[20].

We find high icosahedral symmetry with almost 100% yield near the free energy minima of the icosahedral model clusters. This confirms the reliable formation of near-perfect (anti-)Mackay clusters in the magic number regions[19] (Supplementary Figs. S9 and 10). The occurrence of icosahedral clusters gradually decreases to 0% away from the minima (Fig. 2b). Here, weaker icosahedral symmetry is still present in the form of defected icosahedral clusters (Supplementary Figs. S11 and 12). Furthermore, peaks of icosahedral cluster occurrence are broadened toward smaller system sizes. We believe this broadening appears because icosahedral clusters can accommodate vacancies efficiently, but excess particles must accumulate into defect wedges that break icosahedral symmetry[20]. Decahedral cluster occurrence exclusively peaks at system sizes where decahedral clusters have lower free energies than icosahedral clusters (Fig. 2c). Intriguingly, even in these cases, decahedral clusters remain scarce and occur less than 15% across all system sizes in our tested range. On average, across cluster sizes, we observe only about 3% decahedral clusters, which agrees with the low occurrence of decahedral clusters in experiment (Fig. 1d).

We also measure the presence of icosahedral and decahedral symmetry by point group symmetry quantification[29]. This technique quantifies the presence of point group symmetry in bond-orientational order diagrams. Point group symmetry quantification returns a value of 1 if the bond-orientational order diagram of a given cluster is fully invariant under a point group symmetry and 0 if the bond-orientational order diagram corresponds to a random distribution as found in the fluid state (for details, see "Methods"). Point group symmetry quantification data (open symbols in Fig. 2b, c, Supplementary Fig. S13) closely follows the trend in cluster occurrence obtained from order parameter analysis. This confirms the reliability of our cluster classification. In agreement with the rare observation of cluster with decahedral symmetry, average decahedral point group symmetry remains consistently low.

Our findings show that free energy is not sufficient to explain the cluster occurrence statistics. In particular, it cannot explain the low occurrence of decahedral colloidal clusters. We now analyze simulation trajectories to better understand the role of crystallization kinetics in controlling the cluster polymorph.

## Crystallization pathways of colloidal clusters

We analyze in detail a simulation trajectory forming a decahedral cluster (Fig. 3a–c). The behavior discussed in the following for this trajectory is reproduced in three more independent trajectories, indicating that the exemplary formation pathway is representative of this crystallization process. We find that already before the onset of crystallization, a layered fluid develops with weak icosahedral order near the confinement interface and with an amorphous interior[21,25]. Such fluid pre-ordering resembles the surface-driven crystallization of atomic clusters during quenching[30,31]. Pronounced fluctuation of local order catalyzes the onset of nucleation in the outermost layers[18,30,31]. Initially, five quasi-tetrahedral crystalline grains appear. They are twinned around a common edge that points toward the center of the

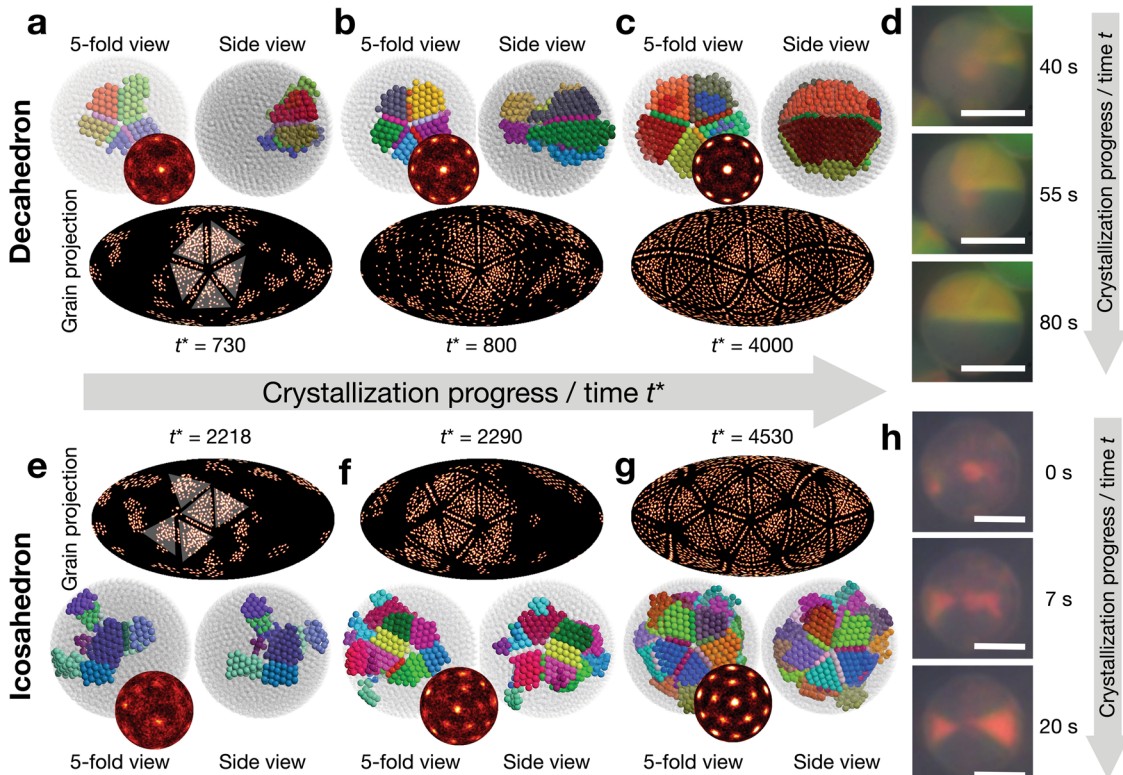

**Fig. 3 | Analysis of kinetic crystallization pathways.** Shown is the crystallization of a decahedral cluster (**a**–**d**) and an icosahedral cluster (**e**–**h**). Simulation trajectories (**a**–**c**, **e**–**g**) are compared to colloid experiments observed with optical microscope (**d**, **h**). **a** Five grains nucleate in a pentagonal bipyramid nucleus. **b**, **c** Grain growth proceeds through the center. **d** In situ optical microscope time series of decahedral cluster formation. **e** Five grains nucleate in a concave nucleus. **f** Crystallization proceeds by the addition of new grains up to 20 grains. **h** In situ optical microscope time series of icosahedral cluster formation. Grains in (**a**–**c**, **e**–**g**) are resolved by a clustering algorithm and colored randomly. Fluid particles and unstable grains are shown as semi-transparent gray. Side view rotated by 90° from a fivefold view. Bond-orientational order diagrams are shown in insets. Grains are highlighted by triangles in grain projection. Scalebars, 5 µm (**d**) and 10 µm (**h**). Note that not all grains in (**d**, **h**) are visible in microscopy but only those that fulfill the Bragg condition.

confinement sphere (Fig. 3a, side view). The five twinned grains form a pentagonal bipyramid nucleus (fivefold view). The bond-orientational order diagram has a sharp central spot surrounded by a ring of ten weak spots (Fig. 3a inset, Supplementary Fig. S14). Its blurred background indicates that most particles remain in the fluid phase.

We also follow the crystallization process by radially mapping the positions of all particles detected to be in crystalline grains onto the confinement sphere and displaying the resulting histogram in Mollweide projection (Fig. 3a, grain projection). Five triangular domains correspond to the five grains of the pentagonal bipyramid. As the simulation proceeds, the cluster becomes more ordered (Fig. 3b, Supplementary Movie 4). The combined evidence of simulation snapshots, bond-orientational order diagrams, and grain projections show that crystallization occurs by grain growth in the direction of the common edge through the cluster center. Grain growth proceeds until the crystal reaches the opposite side of the interface (Fig. 3c).

Crystallization in Fig. 3a–c starts from a single nucleus. We always find this behavior for sufficiently large decahedral clusters. In contrast, for small decahedral clusters, we repeatedly see a different mechanism. Instead of continued growth, we find simultaneous nucleation at antipodes in the interface (Supplementary Fig. S15). Interestingly, the two nuclei have their fivefold axes aligned already at an early stage, even before the grains have grown together. Such an alignment is surprising at first but suggests the presence of strong correlations in the confined fluid. Apparently, the fluctuations in the fluid are strong enough to traverse the entire cluster prior to nucleation.

We visualize a decahedral cluster crystallization process in experiment in an emulsion droplet via the evolution of structural color in real time under an optical microscope (Fig. 3d, Supplementary

Fig. S16, Supplementary Movies 5 and 6)[15]. To slow down droplet drying and ensure sufficient equilibrium time, we immobilize droplets in a sealed polydimethylsiloxane chamber. The sealed droplets dry by diffusion of water from the droplet into the continuous oil phase over the course of three days. During the drying process, colloidal particles in the droplet increase their packing fraction from 1% to approximately 70% and finally consolidate into a crystalline cluster. However, the actual crystallization, judged by the time from the first emergence of structural color until the end of the formation of a complete structural color pattern, occurs much faster in a time scale of only minutes. This finding confirms that crystallization is fast and occurs over less than 1% packing fraction change, justifying the assumption of constant packing fraction in simulation.

Before crystallization starts in a drying droplet, the colloid dispersion remains in the fluid phase. As a result, the droplet appears dimly whitish due to random scattering at the particles. The appearance of a yellow triangular region marks the onset of crystallization. We observe the yellow region to expand from the rim through the droplet center into a semicircle. At that time, no structural color is seen in the lower semicircle. In the upper semicircle, the expanding region with structural color results from a growing crystal grain whose (111) planes align perpendicular to the illumination direction. In this situation, constructively interfered light from the colloidal crystal can be picked up by the objective lens of the microscope[15]. The dark appearance of the lower semicircle indicates that this region also consists of crystal grains, albeit not with (111) planes aligned perpendicular to the illumination direction. Recall that a fluid phase colloid dispersion should appear whitish and milky. Such semicircle structural color pattern has been reported previously by us as unique for

decahedral clusters[15]. Our experimental observations corroborate with the simulations that during the formation of decahedral clusters, multiple grains nucleate near the droplet interface and grow along the direction of common grain edge in the confinement sphere toward the antipode.

We analyze in Fig. 3e–g a simulation trajectory forming an icosahedral cluster to compare it to the case of the decahedral cluster discussed above. The behavior discussed for this trajectory is representative and reproduced in three more independent trajectories. Just like in the trajectory for the decahedral cluster, crystallization begins with the formation of a few quasi-tetrahedral, twinned grains (Fig. 3e, Supplementary Movie 7). However, in contrast to the decahedral trajectory, the grains now form a less compact nucleus of concave shape. We establish the concavity of the nucleus with the bond-orientational order diagram by the absence of some spots in the outer ring and a weaker central spot (Fig. 3e, inset). As crystallization proceeds, the initially nucleated grains do not grow further but retain their size and quasi-tetrahedral shape. Instead, new grains are added one by one, up to eight grains at an intermediate stage (Fig. 3f), and eventually to a complete icosahedron with 20 grains (Fig. 3g, Supplementary Movie 7). Each new grain is twinned with one or several existing grains. This growth mode is also evident in grain projection.

The analysis presented here is for magic number clusters that can form complete icosahedral shells. We confirm that off-magic number clusters with weakly broken icosahedral symmetry[20] follow similar crystallization pathways (Supplementary Fig. S17, Supplementary Movie 8). This demonstrates that the findings in Fig. 3 are independent of the structural quality of the crystalline cluster. Such independence is expected by the following argument. Crystal growth occurs at the intermediate packing fraction when the crystal is weakly in contact with the confinement sphere. At this packing fraction, there does not yet exist a distinction between a magic number cluster and an off-magic number cluster. Only in the final stage of droplet drying do defects first become detectable in the form of a characteristic wedge collecting excess particles[20]. Hence, the formation of defects in the cluster structure occurs at a different stage during cluster formation and is decoupled from the nucleation and growth determining the final symmetry.

Our experimental observation of the crystallization pathway, followed via the evolution of structural color by in situ optical microscopy, corroborates once more the kinetic pathway of simulated icosahedral clusters (Fig. 3h, Supplementary Fig. S18, Supplementary Movies 9 and 10). Icosahedral clusters feature a bow tie structural color motif originating in the projection of two quasi-tetrahedral grains along a twofold axis[15]. Note that similar to the case of the decahedral cluster formation pathway, not all quasi-tetrahedral grains can be simultaneously observed due to the different orientation of grains with respect to the objective lens. The bow tie motif forms from one side of the confinement with the first triangle, i.e., the first grain, and only later grows the second triangle, suggesting new grains appear one by one near existing grains. All experimental observations by optical microscopy are in full agreement with the predictions from the simulation trajectories.

So far, we have shown that crystallization can result in two cluster polymorphs, icosahedral clusters and decahedral clusters, the former occurring more often than thermodynamics predicts. We also contrasted two different nucleus evolution pathways in the crystallization trajectories that arrive at the two cluster symmetries, respectively. Does the difference in crystallization trajectory cause the higher occurrence of icosahedral symmetry? To answer this question, we now pinpoint the precise point when the growing nucleus decides whether to continue to grow into an icosahedral cluster or into a decahedral cluster. We call this point the bifurcation point.

## Bifurcation of the kinetic pathway

We simulate cluster formation with 4399 particles (confinement radius $10.1\sigma$ in Fig. 2). At this system size, defected icosahedral clusters

(indicated by red symbols in Fig. 2) and decahedral clusters (yellow symbols) have similar free energies, and decahedral clusters occur with 10% probability, yet icosahedral symmetry occurs more frequently. From the many simulations performed at this system size, we select a simulation trajectory that results in a decahedral cluster as reference trajectory. Apparently, some rare event occurs along this reference trajectory that is responsible for diverting the simulation away from the typical simulation outcome at this cluster size, an icosahedral cluster.

We analyze the gradual development of decahedral symmetry in the reference trajectory by measuring pressure (Fig. 4a). We follow pressure because it is an indicator of the development of structural order and the appearance of phase transitions. Pressure drops in two stages. The first pressure drop coincides with the appearance of a layered fluid. The second pressure drop coincides with the crystallization[18,19]. Next, we analyze the robustness of the reference trajectory for forming decahedral symmetry by branching off new simulations. A branch-off is performed by initializing a new simulation at a certain branch-off time from the reference trajectory by keeping the particle positions and randomizing all particle velocities. Branch-off simulations are continued for a sufficiently long time until a metastable equilibrium, either a decahedral cluster or an icosahedral cluster, has been formed. Here, we do not distinguish whether icosahedral clusters are defected or not. By performing 120 branch-off simulations and counting the occurrence of icosahedral clusters, a probability for icosahedral symmetry at this branch-off time, called Ih probability, can be determined. Calculating Ih probability along the reference trajectory pinpoints the bifurcation time.

We find that branch-offs at early times, $t^*<680$, predominantly form icosahedral clusters (Ih probability $\sim 90\%$ in Fig. 4a). This means the distribution of simulation outcomes is indistinguishable from crystallizations starting from the melt. For branch-offs at late times, $t^*>750$, decahedral symmetry is unavoidable (Ih probability 0%). At these late times, decahedral symmetry already manifested itself in the reference trajectory. We find that Ih probability rapidly drops from 90 to 0% in a narrow time window and crosses 50% at the bifurcation time $t^*=720$ (Fig. 4a inset). At that time, trajectories bifurcate, and the system switches from a preference for icosahedral symmetry to a preference for decahedral symmetry. Notice that the bifurcation time is located during the early stage of the crystallization process at the beginning of the second pressure drop (gray bar in Fig. 4a).

Close inspection of simulation snapshots near the bifurcation time reveals the structural origin of the bifurcation. When only two or three grains are formed (Fig. 4b), the system follows the icosahedral pathway with high probability. Near the bifurcation time, five grains are present in a pentagonal bipyramid nucleus (Fig. 4c, d). Decahedral symmetry is inevitable only once grain growth reaches the confinement center and grain shapes are more cylindrical than tetrahedral (Fig. 4e, f). These findings demonstrate that the geometry of the growing nucleus and its evolution is crucial for deciding the cluster symmetry.

We find in our simulations that the presence of a pentagonal bipyramid nucleus during the nucleation pathway is a necessary condition for the formation of a decahedral cluster (Figs. 3a, 4d) and that the presence of a concave nucleus during the nucleation pathway is a sufficient condition for the formation of an icosahedral cluster (Fig. 3e). We estimate the probability for forming a pentagonal bipyramid nucleus. Because the layered fluid exhibits icosahedral pre-ordering prior to nucleation[21,22,32], we hypothesize that the pre-ordering templates the fluid into 20 domains that can nucleate grains with low free energy barriers. Recall that nucleation proceeds by sequential addition of grains (this is seen in simulated trajectories and corroborated with experiment observations). We describe the bifurcation process geometrically. The first three grains have unique geometry up to rotation (Fig. 4g). The fourth grain can be added at each of

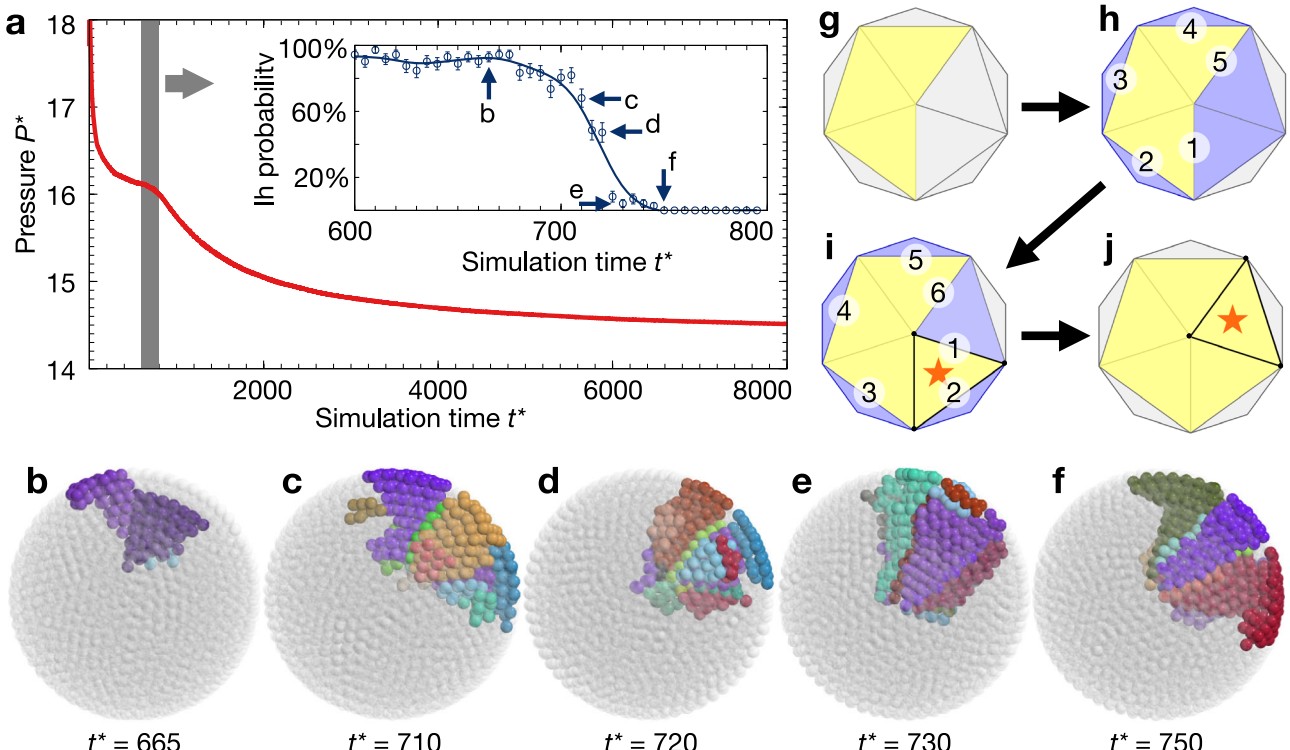

**Fig. 4 | Kinetic bifurcation in simulation trajectories of hard spheres explains the bias toward icosahedral symmetry. a** Dimensionless pressure $P^*$ along a reference trajectory that forms a decahedral cluster with 4399 particles at constant packing fraction 52%. The reference trajectory initializes in a disordered starting configuration. We branch off 120 independent simulations at various simulation times $t^*$ and evaluate their probability of forming an icosahedral cluster (Ih probability). The bifurcation in the inset occurs in a narrow time window highlighted in gray and zoomed in the inset. Error bars indicate short-time averages. **b**–**f** Snapshots of grain development during the formation of the decahedral cluster at the five simulation times indicated by arrows in (**a**). Snapshots are before (**b**, **c**), at (**d**), and after (**e**, **f**) the bifurcation. **g**–**j** A simple geometric argument estimates the formation probability of a pentagonal bipyramid nuclei by assuming random sequential addition of quasi-tetrahedral grains.

five faces (Fig. 4h). Only two choices, face 1 and face 5, proceed toward a pentagonal bipyramid. The fifth grain can likewise be added at six faces (Fig. 4i), two of which, face 1 and face 6, complete the pentagonal bipyramid (Fig. 4j). If each grain addition occurs with equal probability, pentagonal bipyramid formation is predicted for $(2/5) \cdot (2/6) \approx 13\%$ of pathways. This probability of pentagonal bipyramid nuclei is an upper bound for the occurrence of decahedral cluster in the reference trajectory of Fig. 4. Our argument here is certainly simplistic. Nevertheless, it allows us to explain the kinetic bias toward icosahedral symmetry we observe in experiment (Fig. 1d) and simulation (Fig. 2b, c) by the grain evolution during the early stage of the crystallization process.

## Discussion

In conclusion, we revealed an intricate interplay of thermodynamic and kinetic factors during the crystallization of finite hard sphere clusters. Thermodynamics predicts both stable icosahedral clusters and stable decahedral clusters. Kinetics, however, generally favors icosahedral clusters. We explain the bias toward icosahedral symmetry by the shape evolution of the nucleus during the early stage of the crystallization process. Using a geometric argument that estimates the probability of forming a pentagonal bipyramid nucleus, we could rationalize the lower-than-expected occurrence of decahedral clusters.

Our findings are of relevance beyond the colloidal scale. Systems with more complex interactions forming more complex crystal structures commonly have non-classical, multi-step crystallization pathways in which kinetics plays a dominant role. Identifying and characterizing the bifurcations in these pathways will be essential for advancing our understanding of crystallization phenomena.

## Methods

### Particle synthesis

Styrene, acrylic acid, and ammonium peroxodisulfate were purchased from Sigma Aldrich and used as received. Surfactant-free emulsion polymerization was used to produce monodisperse polystyrene (PS) particles with diameter 230 nm and <5% polydispersity[33]. Monodisperse PS with diameter of 500 nm was purchased from microParticle GmbH (Germany).

### Microfluidic fabrication

Microfluidic devices were produced by soft lithography by polydimethylsiloxane (PDMS) replication from master templates[34]. In brief, a thin layer of negative photoresist SU-8 (thickness around 25 μm) was spin-coated on a silicon wafer and patterned with ultraviolet light through a mask of the microfluidic channel design[33]. PDMS (Sylgard 84, Dow Corning) was mixed in a 10:1 ratio with curing agent and poured onto the master substrate to replicate the microfluidic channel structure. After curing at 85 °C in oven overnight, the PDMS chip was peeled off. A 1 mm biopsy punch (Harris Uni-core) was used to create inlets and outlets. After treatment with O₂ plasma for 18 s, the PDMS chip was bound to a pre-cleaned glass slide. Aquapel (PPG Industries) was injected into the microfluidic channel to ensure a hydrophobic surface, which is necessary for device operation.

### Colloidal cluster fabrication

Aqueous dispersion of PS particles of approximately 1 wt% was pumped into the inlet of the microfluidic chip as the dispersed phase. The continuous phase was fluorinated oil (Novec 7500, 3M)

containing 0.1 wt% non-ionic perfluoropolypropyleneglycol-block-polyethyleneglycol-block-perfluoropolypropyleneglycol surfactant[35]. The anionic fluorosurfactant Krytox 157 FSH (Dupont) was also used and produced similar cluster structures of both icosahedral and decahedral symmetries. Microfluidic cells with co-flow microchannel structures (channel width 25 and 50 μm) were used[33]. There are two inlets and one outlet. The aqueous colloidal dispersion phase was introduced to one inlet via HDPE tube connected to a 1 mL syringe. The oil phase was connected by another 1 mL syringe to another inlet. The flow rate of the syringe was controlled by a precision pump (Harvard Apparatus). In the microfluidic channel, the dispersed water phase was injected into a moving continuous oil phase, which breaks up due to water-oil surface tension as discrete water droplets that encapsulate the colloidal particles. Various flow rate ratios (oil and water from 100 μL/h to 1000 μL/h) were used to produce monodisperse droplets of different diameters to screen for colloidal clusters with different system sizes. The droplets were collected in the outlet with excess fluorinated oil in a 1.5 mL glass vial (Carl Roth). The storage glass vial was sealed by parafilm with holes created by a 0.4 mm syringe needle (BD) to slow down the evaporation rate and therefore allow the system to approach thermodynamic equilibrium[19].

## Colloidal cluster characterization

Droplets of the oil containing consolidated colloidal clusters were deposited on a silicon wafer and, after evaporation of the continuous oil phase, observed under a scanning electron microscope (Zeiss Gemini 500). A PDMS chamber with thin glass coverslips (0.17 mm) as observation window were produced, using a piece of silicon wafer as sacrificial spacer. Polyethylene tubes were used to guide the droplets into the chamber through the inlet, which also sealed the chamber for a duration of at least a week. The PDMS chamber was placed under an optical microscope (Leitz, Ergolux) at rest in a bright field in the reflection mode of white light. The water vapor diffusion through the PDMS chamber caused the shrinkage of droplets. A high-resolution CMOS camera (Thorlab) was used to record the evolution of the structural color of drying droplets through 40x and 125x objectives.

## X-ray tomography

A ZEISS Xradia 810 Ultra X-ray microscope equipped with a 5.4 keV rotating anode Cr-source and a Zernike phase ring for positive phase contrast imaging allowed imaging of $16 \times 16$ μm large areas with optical resolutions down to 50 nm (pixel size of 16 nm). Several clusters of one ensemble were prepared on the edge of a sticky carbon pad to find one of the rare decahedron samples by transmission imaging in the microscope. The chosen sample was transferred onto a needle tip in the pre-alignment light microscope of the X-ray microscope for 360° tomography without missing wedge. Some spherical gold nanoparticles deposited on top of the cluster improved image alignment. For the datasets, a tilt series with a total number of 901 transmission images was recorded with an acquisition time of 400 s/frame. The image series was aligned along the rotational axis by the manual tracking of the gold nanoparticles, and the complete two-dimensional dataset was relocated to fit the reconstruction geometry. Acquisition and alignment were performed in the native ZEISS microscope software (XMController and Scout&Scan), and three-dimensional reconstruction was performed employing the Simultaneous Iterative Reconstruction Technique (SIRT) algorithm[36,37] implemented via an in-house Python script based on the Astra-Toolbox. Arivis Vision4D and InViewR were used for visualization, segmentation by virtual reality and quantitative three-dimensional analysis.

## Crystallization of hard sphere clusters

Colloidal clusters were modeled as systems of $N$ identical hard spheres with diameter $\sigma$ and mass $m$ in hard spherical confinement of radius $R$. The evolution of the system was simulated using event-driven molecular dynamics at constant packing fraction $\phi = N(\sigma/(2R))^3 = 52\%$. We chose this packing fraction because it is near the density where nucleation and growth of colloidal clusters are fastest. It is also close to the critical density for solidification. Collision events were sorted in memory by collision time using a priority search tree with $O(1)$ complexity and overlap removal handled in a stable way; pressure was estimated as in prior work[19]. Simulation runs were initialized in a disordered starting configuration or from the particle positions of a prior simulation and continued until the crystallization process was completed, which was detected by the convergence of pressure.

## Numerical grain extraction

Thermal noise of particle motion in the form of vibrations in local cages was reduced prior to structural analysis by short-time averaging of particle positions (Supplementary Fig. S14). Local bond-orientational order parameters[38] $q_{lm}(i) = \sum_{j=1}^{N_b(i)} Y_{lm}(r_{ij})/N_b(i)$ were used to characterize the local environment of all particles $i$, where $N_b(i)$ is the number of nearest neighbor bonds and $Y_{lm}$ the spherical harmonics for the bond vector $r_{ij} = r_j - r_i$. Similar local environments were clustered with the DBSCAN algorithm[39] into crystalline grains. Temporal coherency of grain extraction was guaranteed by pairing a grain in a later simulation snapshot with a grain in an earlier snapshot if both grains share at least half of their particles.

## Characterization of cluster symmetry

Global bond-orientational order parameters[40] were obtained by summing the $q_{lm}$ over all particles, $Q_{lm} = \sum_{i=1}^{N} N_b(i) q_{lm}(i)/\sum_{i=1}^{N} N_b(i)$. Rotationally invariant combinations were constructed as $Q_l = (4\pi \sum_{m=-l}^{l} |Q_{lm}|^2/(2l+1))^{1/2}$. Values for $Q_l$, $k = 4, 6, 8, 12$ were calculated for ideal cluster models (Supplementary Table S1). Given these ideal values and Supplementary Fig. S7, the $Q_6$ order parameter was chosen to distinguish icosahedral and decahedral clusters, first by classifying candidate icosahedral and candidate decahedral clusters and then further by manual analysis.

## Point group symmetry quantification

Besides the use of bond-orientational order parameters, which is a standard technique in the literature, we also analyzed cluster symmetry with the help of a new technique we recently developed called point group symmetry quantification[29]. In point group symmetry quantification, we quantify the presence of point group symmetry in the bond-orientational order diagram by performing a group theoretical analysis of the spherical harmonics expansion with the help of Wigner matrices. We start from point group symmetry order parameters $S_G$ for both point groups $G = I_h$ and $D_h$. Point group order parameters range from 1 (the bond-orientational order diagram is fully symmetric under the chosen point group) to 0 (bond-orientational order is random). The point group symmetry of a given cluster is then quantified by determining the maximum of $S_G$ when scanning over all orientations of the cluster. The occurrence statistics of decahedral and icosahedral clusters and the average point group symmetry were evaluated at each system size (confinement radius) over 192 independent simulations.

## Free energy calculations

Our free energy calculation method[19] was extended to decahedral clusters. The method is based on the Einstein crystal method[41] and implemented in a Monte Carlo simulation framework with harmonic springs coupling particles to a reference configuration. The reference configuration was created by compression to high density to remove noise and subsequent rescaling of the system to the packing fraction of interest. The coupling parameter was varied in logarithmic coordinates over ten orders of magnitude from $10^{-5}$ to $10^5$. Contributions of diffusion to the free energy were sampled efficiently by the introduction of swap moves[42] with nearest neighbors.

## Data availability

All necessary data generated or analyzed during this study are included in this published article and the supporting information. Raw data underlying the figures are available on Zenodo at https://zenodo.org/record/8237372. Other auxiliary data are available from the corresponding authors on request.

## Code availability

Custom event-driven molecular dynamics simulation code used in the current study is available from the corresponding authors on request. Visualization of simulation results and geometric structure analysis was conducted with the software package Injavis, which is available on Zenodo at https://zenodo.org/record/4639570.

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

## Acknowledgements

This research was supported by the Deutsche Forschungsgemeinschaft (DFG, German Research Foundation) Project-ID 416229255—SFB 1411. S.E. acknowledges support by the DFG within the framework of the research training group GRK 1896 (Project-ID 218975129), SFB 1452 (Project-ID 431791331) and the project SP648/8 (Project-ID 316992193). N.V. and M.E. also acknowledge funding by the DFG under projects VO

1824/7-1 and EN 905/2-1, respectively. Computational resources and support provided by the Erlangen Regional Computing Center (RRZE) are gratefully acknowledged.

## Author contributions

The manuscript was written through the contributions of all authors. C.F.M. performed event-driven molecular dynamics simulations and calculated free energies; J.W. fabricated, characterized and monitored the colloidal clusters; S.E. performed X-ray acquisition and analysis; P.B. contributed to grain extraction and cluster classification; N.R.V.-R. quantified point group symmetries; C.F.M., J.W., S.E., N.V., and M.E. wrote the manuscript; E.S., N.V., and M.E. supervised the project; N.V. and M.E. are responsible for the project concept.

## Funding

## Competing interests

The authors declare no competing interests.
