## [Peer Review File · Nature Communications]

REVIEWER COMMENTS

Reviewer #1 (Remarks to the Author):

This is an excellent paper, interesting and clearly written, with significant results on the two different pathways to bulk colloidal crystals with two distinct geometries - icosahedral and decahedral. I recommend publication after the following minor items have been addressed.

Perhaps the terminology "open" nucleus or cluster, in contrast to compact, is standard in the field but it is confusing and mathematically incorrect. The authors seem to mean a cluster with a low coordination number vertex at the center - such as a 4-fold vertex and 4 triangular faces. This is no less compact than a degree 5 or 6 vertex forming a pentagonal or hexagonal domain. It just has a bigger external angle at some point.

In Fig. 1b one sees well-ordered hexagonal patches (blue) but much less ordered pentagonal patches - why is this?

Line 64 should have clusters singular - cluster.

Reviewer #3 (Remarks to the Author):

Report for Early-Stage Bifurcation of Crystallization in a Sphere. by Fru Mwah et al.

The authors build on their previous work which examined the structure of clusters of nanoparticles formed in emulsion droplets and compared the structures formed, determined via tomography, with results from computer simulation. This work takes the approach to a new level, by considering the important question of the kinetic pathway taken by the clusters during self-assembly. The work is interesting, and the use of the simple colloidal model system means that the principles are far-reaching, applicable to very many different materials, such as metal nanoparticles, as the authors make clear. In this respect, the topic is broad and most suitable for a journal such as Nature Communications.

The work appears sound, but before this manuscript can be discussed for publication, there is some quite serious re-writing to be done. I find it pretty much impossible to follow the manuscript in its present form. I think this is because the authors have tried hard to shorten it: lengthening the manuscript to make it more intelligible will help considerably.

Overall I am often confused about what methods the authors are employing, and indeed whether a particular piece of data or analysis pertains to simulation or experiment. This is not helped by the fact that I understand that the tomography can provide data where the particle coordinates are determined, as can the simulations. So please state very clearly whether the data shown in a particular panel is experimental or simulation. Starting with Fig. 1. I suppose that b,c are comparisons of SEMs with some rendered ideal set of coordinates? This is not clearly stated, and it really needs to be, it is very hard to interpret this figure at the moment.

However the issues run rather deeper than this. For example, quite simply I do not understand what is being shown in Fig. 2a. Firstly it is described as a “free energy landscape”. I don’t think it is, in my understanding (based on the book by David Wales “Energy Landscapes”), the (free) energy landscape is the (free) energy of a system as a function of some generalized coordinates for a given system. This on the other hand, appears to be the free energy as a function of the number of particles. More fundamentally, I do not understand what is plotted.

The authors seem to produce the “free energy” from nowhere: firstly, please state in a few lines what is being done here, so that the reader has some chance to understand it. Secondly, when I did look in the Methods section, and understood something of how the authors determine the free energy, I am still none the wiser for what is actually plotted: I understand that an Einstein Crystal method is used for the data: what are the data points, what are the lines, why are they different? What is meant by “simulation”, why is this different from Mackay, anti-Mackay, Decahedron?? I thought that the simulation produced Icosahedral or Decahedral clusters, so why are they different? If the clusters are “poor” and rich in defects then please clarify this.

Many sections of the text are very hard to interpret. For example, I have no idea what “point group symmetry quantification” means. The discussion in lines 138—142 is very vague. So the volume fraction of the droplets increases over time. Why not have a plot showing this? And if it takes 3 days to increase from 1% to 70%, how does this correlate with the crystallisation? At what volume fraction does the system start to crystallize?

Is this work really relevant to the assembly of virus shells? I don’t see why, virus shells are not confined in the manner of the systems here, and also virus shells are not really 3d particles as is considered here. Please drop this analogy as it does not hold.

Figure 3 (like Figs 1 and 2) is hard to interpret. Eventually it seemed that a,b,c and e,f,g are simulation and that d,h are experiment. Please state this in the figure. However, no timescales whatsoever are given in the figure, as far as I can make out. This seems odd, as this figure is so central to the story, which is itself time—dependent. In addition to this, are we to presume that d and h correspond to a,b,c and e,f,g respectively? I find such comparison troubling, as the experimental data in d,h seem to show one and only one ordered region, while the simulated data show rather more of the system being ordered, but that it is comprised of a number of different domains, yet if I understand the experiential data correctly, then there is only one domain?? Or, if this is not the right intrepretation please make the presentation clearer!

Smaller points:

The authors talk of a “critical point” and “critical time”. I would express this differently. In this field, the critical point is a well—defined state point which is not part of this story, and “critical time” is seldom used. I would refer to this as “bifurcation point”.

In short the authors have some interesting data, concerning an important phenomenon. However the manuscript in its present form is very hard to understand, and to follow. I emphasize that what is written above is really only the tip of the iceberg for the revisions that need to be made before the manuscript can be understood sufficiently.

Early-Stage Bifurcation of Crystallization in a Sphere
by C. F. Mbah et al.

This work describes a limited number of experimental results, and a more extensive set of simulations, aimed at elucidating the relative rate of formation of colloidal clusters with different internal structure, using information about the thermodynamics and the kinetics of aggregation of the colloidal spheres confined in evaporating droplets. The experimental system and simulation tools are very close to those described in the authors' previous publications. The main focus of the present work is on the distinction between icosahedral-symmetry crystallites vs. decahedral-symmetry crystallites. While previous work had described varying degrees of perfection of the former, this one recognizes the latter as an alternative mode of crystallization and seeks to shed light on a bifurcation leading crystallization down either path.

The results are visualized impressively and the structural analysis seems sound. In many instances, however, the reader is left wondering whether the data fully supports the conclusions outlined in the introductory paragraph.

1. While the Supplementary Information contains a lot of data on the order parameters used to classify structures, the question remains how the results would change if the criteria are shifted that identify clusters as Ih, Dc, fcc, etc (as in Fig.S8). A bifurcation is only a well-defined event if it separates outcomes of well-defined distinction.
2. This question is very pertinent when comparing the experimental Fig.1d to the theory/simulation figures. The former classifies any cluster as Ih or Dh, with no other possibilities (like defected icosahedral). The caption of Fig.1 also makes it appear as if this experimental classification is done by visual inspection, and thus not by criteria that can be directly translated to those classifying the simulated objects. The numbers of these simulated clusters classified as Ih and Dh, in turn, seem to not always add to 100%. In fact, for many numbers of particles in Fig.2, the majority of clusters seems classified as neither. At the very least one would expect an explicit discussion of these other cluster structures. In the discussion of percentages Fig.2b, Ih is supposed to decay to "0%", but this seems to happen in places where Dh is also very low, so the statement reads misleading.
3. It is not clear from the energy "landscapes" (see a later remark on the use of that term) in Fig.2 whether 192 simulations were conducted for all N , or just those 21 presented in b,c. Whether or not, are the symbols in part a

averages? Are there standard deviations? If the "simulation" results represent good approximations to ground states, how far away are the states obtained from Ih or Dh symmetry from that standard deviation? If "simulation" starts from random configurations and is supposed to find near-ground state energies, why is it beaten by the decahedral minima (and only by those)?

4. Similar questions about the statistics of simulations can be raised about the kinetic part of the investigation. Fig.3 describes "representative" kinetics; while the mechanism and patterns of grain organization are very plausible, there should be some indication in what quantitative sense these cases presented here are representative or typical, and what other, atypical cases would look like. Again, the reader wonders if the classification into Dh and Ih may not be as clear or universal as is suggested in various places of this manuscript.
5. All simulations are conducted at 0.52 volume fraction – why this number? Would the results be different for other (fixed) fractions? In experiment, evaporation densifies the clusters towards crystallization; while this process is carefully controlled to be slow, is it not an important difference to constant-fraction computations? If structural color is the indicator of classification here, and it occurs over a short time at 0.7 volume fraction, why should the 0.52 computations paint an accurate picture?

More minor points:

- The term "energy landscape" is used differently in different fields; in many contexts of particle packings it is meant to indicate different (potential or total) energies that the same set of particles can have, with packing structure making the difference. Here, the "landscape" is different energy values (presumably meant to be ground states) as a function of particle number. While I don't discourage the use of the word entirely, it should be made very clear in the manuscript.
- When the aqueous phase diffuses/evaporates, does it retain capillary bridges inside the cluster? Will this make a difference to the hard-sphere modeling?
- A statement that Ih and Dh free energies are similar at 10.1σ seems to contradict Fig.2a; Dh energy appears significantly lower at that point.
- Beyond basic soft lithography techniques, the microfluidic set-up is never described – are there multiple inlets/outlets for aqueous and oil phases?

In summary, the subject of this work is interesting and a clear description of how crystallization chooses pathways to alternative structures is a valuable goal. A valid explanation should allow for a prediction of a result like Fig.1d, though the binary classification of this figure appears suspect in the light of the rest of the manuscript. In the final part of the paper, a qualitative argument explains that one should expect a lower percentage of Dh structures, but again it is doubtful whether the process of grain assembly described there leads to an unambiguous binary outcome. In its present form, the manuscript does not appear to provide such clarity and predictive power, and instead offers hints towards principles of structural arrangement that might be put in a greater context later, with more quantitative comparisons. Unless the issues mentioned above can be resolved in a substantial revision, I would recommend this paper for publication in a more specialized journal, but not in *Nature Comm.*

Reviewer #1 (Remarks to the Author):

This is an excellent paper, interesting and clearly written, with significant results on the two different pathways to bulk colloidal crystals with two distinct geometries - icosahedral and decahedral. I recommend publication after the following minor items have been addressed.

Our response: We thank the reviewer for the positive comments and helpful suggestions.

Perhaps the terminology "open" nucleus or cluster, in contrast to compact, is standard in the field but it is confusing and mathematically incorrect. The authors seem to mean a cluster with a low coordination number vertex at the center - such as a 4-fold vertex and 4 triangular faces. This is no less compact than a degree 5 or 6 vertex forming a pentagonal or hexagonal domain. It just has a bigger external angle at some point.

Our response: Indeed, the terms "compact nucleus" and "open nucleus" do not follow standard mathematical terminology. We updated the manuscript using the terms "pentagonal bipyramid nucleus" and "concave nucleus" instead.

In Fig. 1b one sees well-ordered hexagonal patches (blue) but much less ordered pentagonal patches - why is this?

Our response: The cluster in Figure 1b has overall icosahedral symmetry. Hexagonal patches correspond to $\{111\}$ planes. These patches are dense and highly ordered crystal planes. What the referee refers to as pentagonal patches are vertex regions of the icosahedral clusters where five crystal grains meet. Each vertex region has at its center a crystallographic defect and as a result is less ordered. We discussed this fact and the reason for the disorder in detail in our previous work, Wang et al. ACS Nano 13, 9005 (2019), <https://doi.org/10.1021/acsnano.9b03039>.

Line 64 should have clusters singular - cluster.

Our response: The typo has been corrected.

Reviewer #2 (Remarks to the Author):

This work describes a limited number of experimental results, and a more extensive set of simulations, aimed at elucidating the relative rate of formation of colloidal clusters with different internal structure, using information about the thermodynamics and the kinetics of aggregation of the colloidal spheres confined in evaporating droplets. The experimental system and simulation tools are very close to those described in the authors' previous publications. The main focus of the present work is on the distinction between icosahedral-symmetry crystallites vs. decahedral-symmetry crystallites. While previous work had described varying degrees of perfection of the former, this one recognizes the latter as an alternative mode of crystallization and seeks to shed light on a bifurcation leading crystallization down either path.

Our response: We thank the reviewer for the detailed comments, which helped to significantly improve the manuscript. Indeed, this manuscript builds on our earlier experimental research. However, there are three aspects, which make this work novel and of high impact:

- (i) This work is the first detailed report (experiment and/or simulation) of decahedral colloidal clusters. Only decahedral nanoclusters have been known to date.
- (ii) This work is distinct from our previous papers because it analyzes colloid crystallization pathways. Our previous papers focus on cluster geometry and structure only.
- (iii) This work contrasts the influence of kinetics in the crystallization of icosahedral and decahedral clusters. We are not aware of such a study in clusters.

The results are visualized impressively and the structural analysis seems sound. In many instances, however, the reader is left wondering whether the data fully supports the conclusions outlined in the introductory paragraph.

We regret that the manuscript did not convey its message convincingly and gave the impression that the results are not fully supported by data. This is certainly not the case. All findings are reliable and robust. Motivated by the comments of the reviewers, we now rewrote all parts of the manuscript to provide better explanations.

The updated abstract makes the following claims, all of which are all well supported:

- Line 12ff: *“We report the observation of decahedral clusters from colloidal particles in emulsion droplets and show that these decahedral clusters can be thermodynamically stable just like icosahedral clusters.”*

We observed decahedral clusters in hundreds of independent experiments and simulations. Some of these are shown in Figure 1. Additional data is found in Supplementary Information. The free energy calculations in Figure 2 are highly precise and capable of distinguishing the relative stability of decahedral and icosahedral colloidal clusters.

- Line 14ff: *“Our hard sphere simulations reveal how the development of the early nucleus shape passes through a bifurcation that decides the cluster symmetry.”*
The bifurcation into decahedral and icosahedral symmetry is detected and characterized in Figure 4 by branching off simulations from a reference trajectory. We note that distinguishing decahedral and icosahedral symmetry in finite clusters is not a challenge for our algorithms (see the detailed discussion below).
- Line 16ff: *“A geometric argument explains why decahedral clusters are kinetically hindered and why icosahedral clusters can be dominant even if they are not the thermodynamic ground state.”* This argument is presented in Figure. 4g-j. It is certainly a simple model. But often simple models are most helpful if they capture the essence of a mechanism.

1. While the Supplementary Information contains a lot of data on the order parameters used to classify structures, the question remains how the results would change if the criteria are shifted that identify clusters as lh, Dc, fcc, etc (as in Fig. S8). A bifurcation is only a well-defined event if it separates outcomes of well-defined distinction.

Our response: The distinction of decahedral symmetry and icosahedral symmetry is not a challenge and poses no real problem. In practice, the binary classification is as easy as telling apart the two bond-orientational order diagrams shown as circular insets in Figure 3c,g and reproduced here on the right. The reason the classification is comparably easy is that icosahedral clusters and decahedral clusters are clearly distinct. They are entirely different crystal structures (think the diamond crystal and fcc) and are not related by small displacements.

The situation is less black and white when distinguishing icosahedral symmetry and defective icosahedral symmetry. The reviewer might have been confused by this distinction. The appearance of defects in icosahedral clusters can be gradual. However, defects in the off-magic number regions are also not a real problem. It is just important to distinguish icosahedral and decahedral symmetry. We write:

Line 261ff: *“The analysis presented here is for magic number clusters that can form complete icosahedral shells. We confirm that off-magic number clusters with weakly broken icosahedral symmetry¹⁹ follow similar crystallization pathways (Supplementary Figure S17, Supplementary Movie 8). This demonstrates that the findings in Figure 3 are independent of the structural quality of the crystalline cluster. Such an independence is expected by the following argument. Crystal growth occurs at intermediate packing fraction when the crystal is weakly in contact with the confinement sphere. At this packing*

fraction, there does not yet exist a distinction between a magic number cluster and an off-magic number cluster. Only in the final stage of droplet drying do defects first become detectable in form of a characteristic wedge collecting excess particles¹⁹. Hence, the formation of defects in the cluster structure occurs at a different stage during cluster formation and is decoupled from the nucleation and growth determining the final symmetry.”

2. This question is very pertinent when comparing the experimental Fig.1d to the theory/simulation figures. The former classifies any cluster as Ih or Dh, with no other possibilities (like defected icosahedral). The caption of Fig.1 also makes it appear as if this experimental classification is done by visual inspection, and thus not by criteria that can be directly translated to those classifying the simulated objects.

Our response: We took great care in the characterization of decahedral clusters. Given experimental limitations, the majority of colloidal clusters were classified from the outside and via structural color patterns. However, we also provide evidence from electron tomography. Tomography extracts the positions of all particles in a colloidal cluster (Supplementary Figure S5). All data fully agrees with the decahedral cluster model. As discussed already above, the classification of decahedral and icosahedral clusters in simulation is highly reliable.

The numbers of these simulated clusters classified as Ih and Dh, in turn, seem to not always add to 100%. In fact, for many numbers of particles in Fig.2, the majority of clusters seems classified as neither. At the very least one would expect an explicit discussion of these other cluster structures. In the discussion of percentages Fig.2b, Ih is supposed to decay to "0%", but this seems to happen in places where Dh is also very low, so the statement reads misleading.

Our response: Indeed, the numbers of clusters identified as Ih and Dh do not add up to 100%. This is because we exclude defected icosahedral clusters from our identification. The existence of these defected icosahedral clusters and details of their structure has been discussed in detail in our previous work. We added the following discussion to the manuscript:

Line 136ff: “We perform the classification for all clusters with decahedral symmetry and high icosahedral symmetry, omitting in our analysis defected cluster with weakly broken icosahedral symmetry. These defected icosahedral clusters dominate in the off-magic number regions, where closed-shell clusters cannot be formed due to accumulation of defects¹⁹.

We find high icosahedral symmetry with almost 100% yield near the free energy minima of the icosahedral model clusters. This confirms reliable formation of near-perfect (anti-)Mackay clusters in the magic number regions¹⁸ (Supplementary Figures S9-10). The occurrence of icosahedral clusters gradually decreases to 0% away from the minima

(Figure 2b). Here, weaker icosahedral symmetry is still present in the form of defected icosahedral clusters (Supplementary Figures S11-12).“

We would like to state once more (see response to point 1 above) that the distinction between icosahedral clusters and defected icosahedral clusters has no effect on the rest of the manuscript. The distinction is merely employed in Figure 2 to correlate the appearance of icosahedral clusters to magic number regions. If we did not make the distinction, the combined icosahedral and defected icosahedral clusters would be present across all cluster sizes. The distinction allows us to correlate the appearance of defects with magic number regions.

3. It is not clear from the energy "landscapes" (see a later remark on the use of that term) in Fig.2 whether 192 simulations were conducted for all N, or just those 21 presented in b,c.

Our response: The 192 simulations were conducted for the 21 cluster sizes shown in Figure 2b,c only. This is stated in the Methods section and now also in the caption of Figure 2:

Line 162: “192 simulations were performed at each confinement radius.“

Whether or not, are the symbols in part a averages? Are there standard deviations?

Our response: The data points in Figure 2a for the simulation results were averaged over five simulations each. We added standard errors to the figure. Data points for free energy calculations for the Mackay, anti-Mackay, and Decahedron model structures are not averaged. We write in the caption of Figure 2:

Line 156ff: “Data points are free energy values ((anti-)Mackay and decahedral clusters) and averaged over five free energy values (simulated clusters; with standard errors).“

If the "simulation" results represent good approximations to ground states, how far away are the states obtained from Ih or Dh symmetry from that standard deviation?

Our response: Free energies for the simulation results form a lower bound of the other free energy curves within the standard deviation. An exception are the decahedral clusters (see the discussion below). The overall low fluctuations and small standard deviations in the free energy data are an indication of the high precision of the free energy calculation presented in this work.

If "simulation" starts from random configurations and is supposed to find near-ground state energies, why is it beaten by the decahedral minima (and only by those)?

Our response: This is an excellent point. The explanation of why it is so difficult to find decahedral clusters in simulation despite their thermodynamic stability (low free energy) is a central point of the manuscript and motivated everything that comes after Figure 2.

We write in the manuscript:

Line 126ff: *“The free energy curve of the simulated clusters (red in Figure 2a) traces the free energy minima of icosahedral model clusters closely but does not entirely trace the free energy minima of decahedral model clusters. Apparently, crystallization in hard sphere simulation consistently fail to reach ground states with decahedral symmetry. This is the case even at size ranges where decahedral clusters are thermodynamically favorable.”*

Why free energy of decahedral clusters is not traced is resolved in the rest of the manuscript.

4. Similar questions about the statistics of simulations can be raised about the kinetic part of the investigation. Fig.3 describes "representative" kinetics; while the mechanism and patterns of grain organization are very plausible, there should be some indication in what quantitative sense these cases presented here are representative or typical, and what other, atypical cases would look like.

Our response: In addition to the data presented in Figure 3, we analyzed three more trajectories each that exhibit the same kinetic behavior. This is now mentioned in the text. We are not aware of or have seen other, atypical cases. We only observe the behavior reported in this manuscript.

Again, the reader wonders if the classification into Dh and Ih may not be as clear or universal as is suggested in various places of this manuscript.

Our response: As discussed in our reply to point 1 above, the classification into Dh and Ih is never problematic.

5. All simulations are conducted at 0.52 volume fraction – why this number? Would the results be different for other (fixed) fractions?

Our response: We now write in the method section:

Line 434ff: *“The evolution of the system was simulated using event-driven molecular dynamics at constant packing fraction $\phi = N(\sigma/(2R))^3 = 52\%$. We chose this packing fraction because it is near the density where nucleation and growth of colloidal clusters is fastest. It is also close to the critical density for solidification.”*

Previous simulations and free energy calculations (Wang et al., Nature Comm. 9, 5259 (2018); <https://doi.org/10.1038/s41467-018-07600-4>) studied other volume fractions and found no significant difference. We expect free energy differences between clusters of different symmetry (Ih and Dh) vary weakly with density. This is a reasonable assumption because the density range where crystallization occurs is small, less than 5%.

In experiment, evaporation densifies the clusters towards crystallization; while this process is carefully controlled to be slow, is it not an important difference to constant-fraction computations? If structural color is the indicator of classification here, and it occurs over a short time at 0.7 volume fraction, why should the 0.52 computations paint an accurate picture?

Our response: This is a misunderstanding. Crystallization does not happen at 0.7 volume fraction. It does, however, happen very fast, i.e. nearly at constant density. We now clarify these aspects better:

Line 212ff: "To slow down droplet drying and ensure sufficient equilibrium time, we immobilize droplets in a sealed polydimethylsiloxane chamber. The sealed droplets dry by diffusion of water from the droplet into the continuous oil phase over the course of three days. During the drying process, colloidal particles in the droplet increase their packing fraction from 1% to approximately 70% and finally consolidate into a crystalline cluster. However, the actual crystallization, judged by the time from the first emergence of structural color until the end of the formation of a complete structural color pattern, occurs much faster in a time scale of only minutes. This finding confirms that crystallization is fast and occurs over less than 1% packing fraction change, justifying the assumption of constant packing fraction in simulation."

More minor points:

The term "energy landscape" is used differently in different fields; in many contexts of particle packings it is meant to indicate different (potential or total) energies that the same set of particles can have, with packing structure making the difference. Here, the "landscape" is different energy values (presumably meant to be ground states) as a function of particle number. While I don't discourage the use of the word entirely, it should be made very clear in the manuscript.

Our response: We removed the term "free energy landscape" from the manuscript.

When the aqueous phase diffuses/evaporates, does it retain capillary bridges inside the cluster? Will this make a difference to the hard-sphere modeling?

Our response: During droplet evaporation (diffusion of water from aqueous droplet to continuous oil phase), the colloidal particles are encapsulated inside the droplet. We can exclude the scenario where colloidal particles stick to the droplet interface, because this would lead to complete coverage of particles at the interface (Pickering emulsion) followed by buckling of the interface as the droplet continues to shrink in volume. We use a fluorinated oil to reduce this affinity of the particles to the interface and thus prevent such buckling.

As the droplet dries, the interface (protected by surfactants at the droplet interface) pushes the particles towards the center. If colloidal particles are surrounded (immersed) in water, there are no

capillary bridges (A). This situation changes once colloidal particles come into contact, which then leads to a resist of compression from the shrinking droplet interface and a deformation of the interface (B). At this point, capillary forces arise. However, the crystallization process is completed while particles are free to move (in scenario A), well before solidification (in scenario B). Therefore, capillary bridges do not influence the crystallization process, and the hard sphere model is suitable to describe the process (<http://doi.org/10.1038/nmat4072>, <http://doi.org/10.1038/s41467-018-07600-4>). Finally, we would like to note that the appearance of structural color, indicative of the formation of a structure, occurs much before the droplet consolidates and form the final, faceted structures, corroborating the proposed mechanism.

A statement that I_h and D_h free energies are similar at 10.1 seems to contradict Fig.2a; D_h energy appears significantly lower at that point.

Our response: We thank the referee for spotting this inaccuracy. We meant to write that the red (simulation) data are similar to the yellow (decahedron) data. The corrected text reads:

Line 290ff: *“We simulate cluster formation with 4399 particles (confinement radius 10.1σ in Figure 2). At this system size, defected icosahedral clusters (indicated by red symbols in Figure 2) and decahedral clusters (yellow symbols) have similar free energies and decahedral clusters occur with 10% probability, yet icosahedral symmetry occurs more frequently.”*

Beyond basic soft lithography techniques, the microfluidic set-up is never described – are there multiple inlets/outlets for aqueous and oil phases?

Our response: A description of the microfluidic setup description as been included in the method section:

Line 393ff: *“Microfluidic cells with co-flow microchannel structures (channel width $25\ \mu\text{m}$ and $50\ \mu\text{m}$) were used³³. There are two inlets and one outlet. The aqueous colloidal dispersion phase was introduced to one inlet via HDPE tube connected to a 1 mL syringe. The oil phase was connected by another 1 mL syringe to another inlet. The flowrate of the syringe was controlled by a precision pump (Harvard Apparatus). In the microfluidic channel, the dispersed water phase was injected into a moving continuous oil phase,*

which breaks up due to water-oil surface tension as discrete water droplets that encapsulate the colloidal particles.”

In summary, the subject of this work is interesting and a clear description of how crystallization chooses pathways to alternative structures is a valuable goal. A valid explanation should allow for a prediction of a result like Fig.1d, though the binary classification of this figure appears suspect in the light of the rest of the manuscript.

Our response: We thank the reviewer for judging our work to be interesting and valuable and the description clear. As discussed in our detailed response, we believe that all our findings are reliable and fully backed up by multiple techniques and measurements (synthesis, tomography, simulation, free energy). All theoretical findings fully agree with experimental results.

In the final part of the paper, a qualitative argument explains that one should expect a lower percentage of Dh structures, but again it is doubtful whether the process of grain assembly described there leads to an unambiguous binary outcome.

Our response: Our experimental data leaves no doubt about the existence of decahedral clusters. It is no challenge to distinguish clusters with decahedral symmetry from clusters with icosahedral symmetry in simulation (see above).

In its present form, the manuscript does not appear to provide such clarity and predictive power, and instead offers hints towards principles of structural arrangement that might be put in a greater context later, with more quantitative comparisons. Unless the issues mentioned above can be resolved in a substantial revision, I would recommend this paper for publication in a more specialized journal, but not in Nature Comm.

Our response: We addressed all questions and concerns of the Reviewer in this substantial revision. We believe all outstanding issues have been resolved and hope the Reviewer can now recommend the manuscript for publication in *Nature Communications*.

Reviewer #3 (Remarks to the Author):

The authors build on their previous work which examined the structure of clusters of nanoparticles formed in emulsion droplets and compared the structures formed, determined via tomography, with results from computer simulation. This work takes the approach to a new level, by considering the important question of the kinetic pathway taken by the clusters during self-assembly. The work is interesting, and the use of the simple colloidal model system means that the principles are far-reaching, applicable to very many different materials, such as metal nanoparticles, as the authors make clear. In this respect, the topic is broad and most suitable for a journal such as Nature Communications.

Our response: We thank the reviewer for the detailed comments and suggestions, which helped significantly to improve our manuscript.

The work appears sound, but before this manuscript can be discussed of publication, there is some quite serious rewriting to be done. I find it pretty much impossible to follow the manuscript in its present form. I think this is because the authors have tried hard to shorten it: lengthening the manuscript to make it more intelligible will help considerably.

Our response: We realize that the explanations in the manuscript were not as clear as they should have been to convey our findings convincingly. The reason was, as the reviewer suspected, a too strong shortening. Motivated by the Reviewer's comments, we rewrote the entire manuscript. In this process, the main text (up to end of Discussion) was significantly expanded by more than 40% from less than 9 pages to nearly 13 pages. Many additional explanations and logical connections were added as clarifications. Our focus was to make the manuscript more accessible to a broader readership and more intelligible.

Overall I am often confused about what methods the authors are employing, and indeed whether a particular piece of data or analysis pertains to simulation or experiment. This is not helped by the fact that I understand that the tomography can provide data where the particle coordinates are determined, as can the simulations. So please state very clearly whether the data shown in a particular panel is experimental or simulation. Starting with Fig. 1. I suppose that b,c are comparisons of SEMs with some rendered ideal set of coordinates? This is not clearly stated, and it really needs to be, it is very hard to interpret this figure at the moment.

Our response: Indeed, Figure 1 contains experimental results only with the exception of Figure 1b,c, which as the reviewer correctly states are ideal geometric models. Figure 2 contains numerical/simulation data only. Figure 3 contains simulation data with the exception of Figure 3d,f, which are from optical microscopy. Figure 4 contains simulation data only and a theoretical model. We

now state in each figure caption clearly and throughout the text consistently from which method or with technique each data was obtained.

However the issues run rather deeper than this. For example, quite simply I do not understand what is being shown in Fig. 2a. Firstly it is described as an “free energy landscape”. I don’t think it is, in my understanding (based on the book by David Wales “Energy Landscapes”), the (free) energy landscape is the (free) energy of a system as a function of some generalized coordinates for a given system. This on the other hand, appears to be the free energy as a function of the number of particles. More fundamentally, I do not understand what is plotted.

Our response: We removed the term “free energy landscape”. Calculating free energy is highly valuable for us because it allows us to identify magic number regions and compare the relative stability of icosahedral and decahedral colloidal clusters. It also helps us to understand whether and how closely simulations follow constructed, ideal cluster models.

The authors seem to produce the “free energy” from nowhere: firstly, please state in a few lines what is being done here, so that the reader has some chance to understand it. Secondly, when I did look in the Methods section, and understood something of how the authors determine the free energy, I am still none the wiser for what is actually plotted: I understand that an Einstein Crystal method is used for the data:

Our response: We are currently preparing a separate manuscript explaining the details of the free energy calculation technique. The free energy calculation technique has already been used in our previous work where it has also been discussed in more detail [Wang et al., “Magic Number Colloidal Clusters as Minimum Free Energy Structures”, Nature Comm. 9, 5259 (2018); <https://doi.org/10.1038/s41467-018-07600-4>] and applied with much success. We can compare across particle numbers by subtracting a smooth background (F_0) to extract free energy fluctuations. This allows us to identify magic number regions. The present manuscript now contains a brief but complete description of all necessary steps and details in the method section. We write:

Line 109ff: “To understand the relative thermodynamic stability of icosahedral and decahedral clusters, we calculate free energies with high precision using a simulation framework with harmonic springs and Monte Carlo swaps of spring developed in our previous work¹⁸ (see also Methods).”

And in Methods we write:

Line 471ff: “Our free energy calculation method¹⁸ was extended to decahedral clusters. The method is based on the Einstein crystal method³⁹ and implemented in a Monte Carlo simulation framework with harmonic springs coupling particles to a reference configuration. The reference configuration was created by compression to high density to remove noise and subsequent rescaling of the system to the packing fraction of interest.

The coupling parameter was varied in logarithmic coordinates over ten orders of magnitude from 10^{-5} to 10^5 . Contributions of diffusion to the free energy was sampled efficiently by the introduction of swap moves⁴⁰ with nearest neighbors.”

what are the data points, what are the lines, why are they different?

Our response: We added the following text to the caption of Figure 2:

Line 156ff: “Data points are free energy values ((anti-)Mackay and decahedral clusters) and averaged over five free energy values (simulated clusters; with standard errors). Lines are guides to the eyes.”

What is meant by “simulation”, why is this different from Mackay, anti-Mackay, Decahedron?? I thought that the simulation produced Icosahedral or Decahedral clusters, so why are they different?

Our response: We discuss these questions in the updated manuscript text:

Line 112ff: “Calculations are performed for ideal icosahedral model clusters (blue in Figure 2a, including both Mackay and anti-Mackay type, Supplementary Figure S6) and ideal decahedral model clusters (yellow).“

And further down we write:

Line 123ff: “We also perform free energy calculations for hard sphere clusters that were crystallized from the melt in event-driven molecular dynamics simulations at constant packing fraction. These clusters, termed ‘simulated clusters’, constitute our closest computational analogues to the experimental colloidal clusters.“

Simulations do produce icosahedral and decahedral clusters. But only rarely produce decahedral clusters and often the icosahedral clusters obtained contain defects. This is why a comparison to ideal model clusters is helpful.

If the clusters are “poor” and rich in defects then please clarify this.

Our response: We distinguish icosahedral clusters and defected icosahedral clusters. This is now clarified better in the text:

Line 136ff: “We perform the classification for all clusters with decahedral symmetry and high icosahedral symmetry, omitting in our analysis defected cluster with weakly broken icosahedral symmetry. These defected icosahedral clusters dominate in the off-magic number regions, where closed-shell clusters cannot be formed due to accumulation of defects¹⁹.”

Many sections of the text are very hard to interpret. For example, I have no idea what “point group symmetry quantification” means. The discussion in lines 138—142 is very vague.

Our response: Point group symmetry quantification is a method to characterize the point symmetry of a cluster in a quantitative way. We developed this technique, in part specifically for this manuscript. Details are found in this preprint: <https://arxiv.org/abs/2106.14846> (Ref. 28). Point group symmetry quantification complements measurements from ordered parameters (which is a more traditional way to characterize cluster structure). The update manuscript text now reads:

Line 167ff: *“We also measure the presence of icosahedral and decahedral symmetry by point group symmetry quantification²⁸. This technique quantifies the presence of point group symmetry in bond-orientational order diagrams. Point group symmetry quantification returns a value of 1 if the bond-orientational order diagram of a given cluster is fully invariant under a point group symmetry and 0 if the bond-orientational order diagram corresponds to a random distribution as found in the fluid state (for details see Methods).”*

A new section has been added to Methods:

Line 458ff: *“**Point group symmetry quantification.** Besides the use of bond-orientational order parameters, which is a standard technique in the literature, we also analyzed cluster symmetry with the help of a new technique we recently developed called point group symmetry quantification²⁸. In point group symmetry quantification, we quantify the presence of point group symmetry in the bond-orientational order diagram by performing a group theoretical analysis of the spherical harmonics expansion with the help of Wigner matrices. We start from point group symmetry order parameters S_G for both point groups $G = I_h$ and D_h . Point group order parameters range from 1 (the bond-orientational order diagram is fully symmetric under the chosen point group) to 0 (bond-orientational order is random). The point group symmetry of a given clusters is then quantified by determining the maximum of S_G when scanning over all orientations of the cluster. The occurrence statistics of decahedral and icosahedral clusters and the average point group symmetry were evaluated at each system size (confinement radius) over 192 independent simulations.“*

So the volume fraction of the droplets increases over time. Why not have a plot showing this? And if it takes 3 days to increase from 1% to 70%, how does this correlate with the crystallisation? At what volume fraction does the system start to crystallize?

Our response: The volume fraction where crystallization occurs cannot easily be estimated from our experiments with high precision. For this it would be necessary to measure the fluid volume in the droplet during evaporation with sufficient precision. This is at present not possible beyond a very rough approximation. It is certainly an interesting point. We plan to address it in a future work.

We also would like to note that the overall time scale in experiment is not very relevant since the critical time of the actual crystallization process is in the order of minutes (up to an hour; see Figure 3), which is very short compared to the full drying process (up to days). We write:

Line 215ff: *“During the drying process, colloidal particles in the droplet increase their packing fraction from 1% to approximately 70% and finally consolidate into a crystalline cluster. However, the actual crystallization, judged by the time from the first emergence of structural color until the end of the formation of a complete structural color pattern, occurs much faster in a time scale of only minutes. This finding confirms that crystallization is fast and occurs over less than 1% packing fraction change, justifying the assumption of constant packing fraction in simulation.”*

Furthermore, we could even slow down the crystallization process manually further (but are not currently doing so), so that in the critical volume fraction there is sufficient time for ordering to occur. In principle, the drying process can be sped up a lot when the droplet increases from 15% to ~40%, then slowed down for the phase transition, and then sped up again afterwards.

Is this work really relevant to the assembly of virus shells? I don't see why, virus shells are not confined in the manner of the systems here, and also virus shells are not really 3d particles as is considered here. Please drop this analogy as it does not hold.

Our response: The analogy to virus shells has been dropped.

Figure 3 (like Figs 1 and 2) is hard to interpret. Eventually it seemed that a,b,c and e,f,g are simulation and that d,h are experiment. Please state this in the figure.

Our response: The following text has been added to the figure caption of Figure 3:

Line 239ff: *“Simulation trajectories (a-c, e-g) are compared to optical microscopy experiments (d,h).”*

However, no timescales whatsoever are given in the figure, as far as I can make out. This seems odd, as this figure is so central to the story, which is itself time-dependent.

Our response: Times for the simulations (in dimensionless units, t^*) as well as for experiment (in seconds, t) have now been included in an updated Figure 3.

In addition to this, are we to presume that d and h correspond to a,b,c and e,f,g respectively? I find such comparison troubling, as the experimental data in d,h seem to show one and only one ordered region, while the simulated data show rather more of the system being ordered, but that it is comprised of a number of different domains, yet if I understand the experiential data correctly, then there is only one domain?? Or, if this is not the right interpretation please make the presentation clearer!

Our response: No, there is not only one grain. It is simply the case that not all grains are visible as the visibility of a grain requires the Bragg condition to be fulfilled for this grain. This fact is the main focus of our earlier paper, Ref. 24 [Wang et al., Adv. Functional Mater 30, 1907730 (2020); <https://doi.org/10.1002/adfm.201907730>]. In this reference we discuss in detail how optical microscopy confirms the cluster symmetry and closely follows the propagation of the crystallization front. We added a note to the caption of Figure 2:

Line 246ff: *“Note that not all grains in (d,h) are visible in microscopy but only those that fulfill the Bragg condition.”*

We also added additional explanations to the text:

Line 224ff: *“We observe the yellow region to expand from the rim through the droplet center into a semicircle. At that time no structural color is seen in the lower semicircle. In the upper semicircle, the expanding region with structural color results from a growing crystal grain whose {111} planes align perpendicular to the illumination direction. In this situation, constructively interfered light from the colloidal crystal can be picked up by the objective lens of the microscope.¹⁵ The dark appearance of the lower semicircle indicates that this region also consists of crystal grains, albeit not with {111} planes aligned perpendicular to the illumination direction. Recall that a fluid phase colloid dispersion should appear whitish and milky. Such semicircle structural color pattern has been reported previously by us as unique for decahedral clusters.¹⁵”*

Smaller points: The authors talk of a “critical point” and “critical time”. I would express this differently. In this field, the critical point is a well—defined state point which is not part of this story, and “critical time” is seldom used. I would refer to this as “bifurcation point”.

Our response: We replaced “critical point” by “bifurcation point” and “critical time” by “bifurcation time”.

In short the authors have some interesting data, concerning an important phenomenon. However the manuscript in its present form is very hard to understand, and to follow. I emphasize that what is written above is really only the tip of the iceberg for the revisions that need to be made before the manuscript can be understood sufficiently.

Our response: We would like to thank the reviewer again for the detailed comments and suggestion. These comments and suggestions are much appreciated and helped significantly to improve the paper and to make it accessible to the broad audience of *Nature Communications*. Let us finally note that the edits we directly discussed above are also only the tip of the iceberg for revisions we performed. We essentially entirely rewrote the manuscript. We hope the reviewer now finds our presentation convincing and clear. We hope the manuscript is now ready for publication.

REVIEWER COMMENTS

Reviewer #1 (Remarks to the Author):

I am happy with the revisions and recommend publication

Reviewer #3 (Remarks to the Author):

Report for Fru Mbah et al.; “Early-Stage Bifurcation of Crystallization in a Sphere”.

The authors present a most interesting development of their previous work [Nature Commun 9, 5259 (2018)] in their model system of colloidal particles confined within emulsion droplets. Previously, they considered the final state and considered the geometric properties of the large colloidal clusters that assembled in this geometry. In the present manuscript, they have gone very much further, because they have considered the time—evolution of the system and in doing so have discerned a very intriguing bifurcation phenomenon. Using a combination of experiment and computer simulation, they analyze this bifurcation behavior and arrive at a compelling geometric argument for the formation of decahedral (rarely) and icosahedral (usually) structures. The authors rightly point out that their basic colloidal model system is applicable to a wide range of materials and thus is appropriate for a wide readership as is the case for Nature Commun. I think this is an important and general development and should enjoy considerable attention and thus I am strongly supportive of its publication in Nature Commun. In fact, despite the wide use of these colloidal model systems to understand general processes in condensed matter, I am unaware of any similar experimental paper which addresses such a bifurcation, or indeed selection of polymorphic routes in assembly, as the authors consider here. This lack of any precedent in my view is strong evidence for ensuring this work reaches a wide audience.

That said, I do have some concerns which the authors should address:

(1) This might seem a little semantic, but isn't “crystallization” a bulk phenomenon? If the term is used in the cluster literature, so be it, but it would seem to me that “assembly” or “ordering within a cluster” might be more appropriate.

(2) As noted above, I am unaware of published experimental work addressing this kind of phenomenon, but there is simulation work on hard spheres that addresses polymorphism which I think that the authors should cite: Leoni and Russo, Phys. Rev. X, 2021, 11, 031006.

(3) Do the authors think hydrodynamic interactions could be important and if not why not?

(4) I think that the authors could do more to emphasize the generality of their approach. What materials might hard spheres be used to address in particular? Are there specific predictions that could be made from these methods?

smaller points:

(5) line 29 — should read “face centered cubic”

(6) line 95 — what exactly is used to explore the cluster geometry? This sentence is unclear.

(7) line 314 — is it not $t^* > 750$, not $t^* < 750$ as written??

Early-Stage Bifurcation of Crystallization in a Sphere (revised)
by C. F. Mbah et al.

The referee appreciates the extensive re-writing of this manuscript as compared to its initial version. The text is certainly more easily understood now and the explanations are more coherent, particularly pertaining to the different measures used to quantify simulated structures. Some figures could have similarly benefited from more extensive revision, though improvements as in Fig.2a are acknowledged, and the more detailed explanations help understanding of the figures as well.

The revised version is also more successful in stating its main results and conclusions. However, a few issues remain for a reader not intimately familiar with the subject:

1. It is curious that Fig.1d contains the only quantitative experimental data in the paper, and is hardly referenced in the text. The only qualitative statement drawn from it is that Dh clusters are significantly less frequent than Ih clusters. Why is this never quantified? What absolute numbers do the yellow and blue bars represent? Why are the occurrence percentages not compared to at least the estimate of the combinatorial kinetic-pathway argument near the end of the manuscript? This qualitative statement appears to be the motivation for the entire manuscript, i.e., (i) Dh clusters were found, (ii) they are rare – why?
2. One may also have expected a comparison between these Fig.1d data and the classification results of Fig.2bc, but there is none. There might be several reasons for this: experimental clusters are binarily classified, while simulated clusters have defected Ih as a possible classification. Nevertheless, cannot the Dh percentages be compared? Also, the experimental clusters seem to be typically much larger than the simulated ones, and there is a marked difference in Dh occurrence between smaller experimental clusters and larger ones (if I am interpreting the 10k and 35k particle numbers correctly, which are, again, not discussed). Are the differences in cluster size an obstacle to comparison? If so, why? Is the apparent trend of higher Dh percentage for larger clusters reflected in the simulations? Can it be rationalized?
3. Maybe a more minor point, but "thermodynamically stable" and "thermodynamically favorable" seem to be used almost interchangeably. Is it clear that the free-energy simulations provide thermodynamically stable (ground) states? Apparently not always, as in some cases the idealized decahedron is lower in energy. But then, how do we know that those idealized states are indeed ground states for a given number of particles? They are

favorable over Ih states (and the simulation outcomes), but do we know they are stable?

4. 1.314/315 This should probably read $t^* > 750$?

I reiterate that this is very interesting work of potential benefit to a wide audience of researchers. This version of the manuscript is improved, but the remaining points detailed above need to be substantially addressed for it to make a real impact. Centrally, I do not understand why this manuscript skirts around the question of even semi-quantitatively comparing its central outcome (from simulation) to its central experimental finding, nor why this finding is so fleetingly treated in one sub-figure. If there are reasons for why such a comparison makes no sense or should be approached with caution, such reasons should be stated clearly.

Reviewer #1 (Remarks to the Author):

I am happy with the revisions and recommend publication.

Our response: We thank the reviewer for the recommendation for publication.

Reviewer #2 (Remarks to the Author):

The referee appreciates the extensive re-writing of this manuscript as compared to its initial version. The text is certainly more easily understood now and the explanations are more coherent, particularly pertaining to the different measures used to quantify simulated structures. Some figures could have similarly benefited from more extensive revision, though improvements as in Fig.2a are acknowledged, and the more detailed explanations help understanding of the figures as well.

Our response: We thank the reviewer for the overall positive and detailed comments that helped further improve the manuscript.

The revised version is also more successful in stating its main results and conclusions. However, a few issues remain for a reader not intimately familiar with the subject:

1. It is curious that Fig.1d contains the only quantitative experimental data in the paper, and is hardly referenced in the text. The only qualitative statement drawn from it is that Dh clusters are significantly less frequent than lh clusters. Why is this never quantified? What absolute numbers do the yellow and blue bars represent?

Our response: Indeed, we already provided quantitative data but did not list the absolute numbers. They are now stated in the caption of Figure 1 at lines 110-111:

“A total number of 100 (68) clusters, among them 2 (11) decahedral clusters (yellow color), were analyzed for clusters with particle number 10k (35k).“

The relative numbers are mentioned in the text at lines 82-84:

“While colloidal clusters of both symmetries are present across all cluster sizes, the occurrence of decahedral clusters is generally much lower with $(2.0 \pm 1.4)\%$ decahedral cluster for clusters with approximately 10000 particles and $(16.2 \pm 4.5)\%$ decahedral clusters for clusters with approximately 35000 particles (Figure 1d).“

Why are the occurrence percentages not compared to at least the estimate of the combinatorial kinetic-pathway argument near the end of the manuscript? This qualitative statement appears to be the motivation for the entire manuscript, i.e., (i) Dh clusters were found, (ii) they are rare – why?

Our response: Indeed, the unexpectedly low occurrence of Dh clusters compared to Ic clusters is a central motivation for the manuscript. We clearly demonstrate this low occurrence in Figure 1d and then, as a function of particle number, in Figure 2. We already made a direct comparison between theory and experiment as requested by the reviewer. The referee might have missed it because it is the last sentence before the discussion starts. Lines 368-370:

“Our argument here is certainly simplistic. Nevertheless, it allows us to explain the kinetic bias towards icosahedral symmetry we observe in experiment (Figure 1d) and simulation (Figure 2b-c) by the grain evolution during the early stage of the crystallization process.”

Note that analyzing the cluster type in experiment is laborious and was not a central point of the manuscript. All we wanted to do is to show that (1) decahedral clusters exist (i.e. their number is not zero); and (2) decahedral clusters occur much less frequently than icosahedral clusters. Figure 1d clearly accomplishes both these goals. A significantly more comprehensive analysis of the occurrence of icosahedral, decahedral and face-centered cubic clusters across cluster size will require a separate manuscript. The focus in the present paper was on the mechanism, i.e. the bifurcation.

2. One may also have expected a comparison between these Fig.1d data and the classification results of Fig.2bc, but there is none. There might be several reasons for this: experimental clusters are binarily classified, while simulated clusters have detected Ih as a possible classification. Nevertheless, cannot the Dh percentages be compared?

Our response: We now make this comparison quantitative and provide numbers. Lines 160-162:

“Intriguingly, even in these cases decahedral clusters remain scarce and occur less than 15% across all system sizes in our tested range. In average across cluster sizes, we observe only about 3% decahedral clusters, which agrees with the low occurrence of decahedral clusters in experiment (Figure 1d).”

Within the accuracy of our analysis, which is not very high (see the discussion above), experiment and simulation agree. The binary classification is not an issue because defected Ih clusters are structurally clearly distinct from Dh clusters and in comparison much more similar to non-defected Ih cluster. In fact, experimentally it is difficult to distinguish defected Ih cluster from non-defected ones. The only way to make such a distinction is to perform an electron tomography study on an individual cluster, as we previously reported in Ref. [20]. Additionally, both defected and non-defected Ih clusters follow nearly identical formation pathways (SI Movies 7, 8) and these formation pathways are in stark contrast to the formation pathway of a Dh cluster.

Also, the experimental clusters seem to be typically much larger than the simulated ones, and there is a marked difference in Dh occurrence between smaller experimental clusters and larger ones (if I am

interpreting the 10k and 35k particle numbers correctly, which are, again, not discussed). Are the differences in cluster size an obstacle to comparison? If so, why?

Our response: Indeed, the experimental clusters are larger. We did not study larger clusters in simulation or with free energy calculations because we wanted to demonstrate that decahedral clusters are prevalent even for small cluster sizes were previously (e.g. <https://doi.org/10.1038/nmat4072>) icosahedral clusters were reported exclusively. We do not believe the cluster size difference is a problem. Explaining the less frequent observation of decahedral clusters is certainly possible with our available data.

Is the apparent trend of higher Dh percentage for larger clusters reflected in the simulations? Can it be rationalized?

Our response: Indeed, we observe in simulations a trend that decahedral clusters become more frequent towards larger systems sizes, just like in experiment (Figure 1d). But we do not wish to study this trend systematically here. A follow-up work will show how towards larger cluster sizes Ih clusters decrease in occurrence and Dh clusters (but importantly also fcc clusters) increase in occurrence.

3. Maybe a more minor point, but “thermodynamically stable” and “thermodynamically favorable” seem to be used almost interchangeably.

Our response: We replaced “thermodynamically favorable” by “thermodynamically stable” throughout the manuscript to avoid confusion.

Is it clear that the free-energy simulations provide thermodynamically stable (ground) states? Apparently not always, as in some cases the idealized decahedron is lower in energy. But then, how do we know that those idealized states are indeed ground states for a given number of particles? They are favorable over Ih states (and the simulation outcomes), but do we know they are stable?

Our response: We are not sure we understand this comment correctly. The red data points in Figure 2a are from long simulations, which aim to reproduce the experiment. Blue (yellow) data points are from constructed icosahedral (decahedral) clusters, which were simulated long enough to reach a local free energy minimum, but not long enough for large-scale structural reorganizations (e.g. decahedral to icosahedral, or icosahedral to defected icosahedral). A main finding of this figure is that our simulations sometimes do not reach decahedral clusters, even if those clusters have the lowest free energy. This demonstrates that indeed, simulations can have troubles reaching a global free energy minimum (or ground state). The fact that we observe both icosahedral clusters and decahedral clusters at the same particle number (Figure 1d) confirms that experiment also does not reliably reach the ground state.

Explaining why it is difficult to reach the ground state is, as the reviewer knows, the central theme of our manuscript. We do not require the idealized (or constructed) states to be ground states.

4. I.314/315 This should probably read $t^* > 750$?

Our response: This was a typo. It has been corrected.

I reiterate that this is very interesting work of potential benefit to a wide audience of researchers. This version of the manuscript is improved, but the remaining points detailed above need to be substantially addressed for it to make a real impact. Centrally, I do not understand why this manuscript skirts around the question of even semi-quantitatively comparing its central outcome (from simulation) to its central experimental finding, nor why this finding is so fleetingly treated in one sub-figure. If there are reasons for why such a comparison makes no sense or should be approached with caution, such reasons should be stated clearly.

Our response: We thank the reviewer again for the positive words. We replied to the reviewer's comments in detail in this rebuttal and believe we addressed all points. There is no reason a semi-quantitative comparison should be avoided. We believed we already made such a comparison in the last version, but maybe it was not clear enough. We added explanations and absolute numbers to make this comparison clearer, which is certainly helpful to the readers and we are grateful for these suggestions.

Reviewer #3 (Remarks to the Author):

Report for Fru Mbah et al.; "Early-Stage Bifurcation of Crystallization in a Sphere".

The authors present a most interesting development of their previous work [Nature Commun 9, 5259 (2018)] in their model system of colloidal particles confined within emulsion droplets. Previously, they considered the final state and considered the geometric properties of the large colloidal clusters that assembled in this geometry. In the present manuscript, they have gone very much further, because they have considered the time—evolution of the system and in doing so have discerned a very intriguing bifurcation phenomenon. Using a combination of experiment and computer simulation, they analyze this bifurcation behavior and arrive at a compelling geometric argument for the formation of decahedral (rarely) and icosahedral (usually) structures. The authors rightly point out that their basic colloidal model system is applicable to a wide range of materials and thus is appropriate for a wide readership as is the case for Nature Commun. I think this is an important and general development and should enjoy considerable attention and thus I am strongly supportive of its publication in Nature Commun. In fact, despite the wide use of these colloidal model systems to understand general processes in condensed matter, I am unaware of any similar experimental paper which addresses such

a bifurcation, or indeed selection of polymorphic routes in assembly, as the authors consider here. This lack of any precedent in my view is strong evidence for ensuring this work reaches a wide audience.

Our response: We thank the reviewer for the positive comments and efforts put into providing helpful comments and suggestions.

That said, I do have some concerns which the authors should address:

(1) This might seem a little semantic, but isn't "crystallization" a bulk phenomenon? If the term is used in the cluster literature, so be it, but it would seem to me that "assembly" or "ordering within a cluster" might be more appropriate.

Our response: The term "crystallization" fits our system. First, crystallization generally occurs in two steps, nucleation and growth. Our clusters show defined nucleation with critical nucleus size smaller than the confinement volume (Figure 4d). Second, small nanocrystals contain similar numbers of particles to our clusters and are described in the literature to form by crystallization (sometimes termed "nanocrystallization"). Third, the term crystallization is indeed used universally in the literature for "processes over length scales ranging from the atomic to hundreds of micrometers, and to originate from a wide range of mechanisms", as stated in <https://doi.org/10.1002/adma.202001068>.

(2) As noted above, I am unaware of published experimental work addressing this kind of phenomenon, but there is simulation work on hard spheres that addresses polymorphism which I think that the authors should cite: Leoni and Russo, Phys. Rev. X, 2021, 11, 031006.

Our response: We thank the reviewer for suggesting this paper and added it as Ref. [17] in lines 56-58:

"And while the phase behavior of hard spheres is dominated by entropy¹⁶, it retains complex nucleation pathways including polymorphism¹⁷."

(3) Do the authors think hydrodynamic interactions could be important and if not why not?

Our response: We do not expect hydrodynamic interactions to be crucial for this work. Hydrodynamic interactions affect crystallization speed [Radu, Schilling, Europhys. Lett. 105, 26001 (2014); Roehm, Kesselheim, Arnold, Soft Matter 10, 5503 (2014)] and colloidal aggregation far from equilibrium [Furukawa, Tanaka, PRL 104, 1 (2010); de Graaf, Poon, Haughey, Hermes, Soft Matter 15, 10 (2019)] but has a weak influence on the equilibrium cluster structure and near-equilibrium structure formation.

(4) I think that the authors could do more to emphasize the generality of their approach. What materials might hard spheres be used to address in particular? Are there specific predictions that could be made from these methods?

Our response: As in all hard sphere studies, our simulations have no particular material in mind. Rather, hard spheres are used as a model system of colloidal particles with generic, dominantly entropic interactions (see Ref. [16]). Our main motivation was to understand the relative thermodynamic stability of icosahedral and decahedral clusters and to resolve kinetic formation pathways of these clusters. Our predictions can be compared to other systems that show icosahedral and decahedral twinning, for example twinned nanoparticles, which are not synthesized routinely. While here we did not investigate how to control such kinetic bifurcation, such control may be helpful to guide synthesis and other assembly methods towards a desired cluster type.

smaller points:

(5) line 29 — should read “face centered cubic”

Our response: This typo has been corrected.

(6) line 95 — what exactly is used to explore the cluster geometry? This sentence is unclear.

Our response: The sentence has been corrected. It now reads:

“We use particle simulations to explore the cluster geometry [...]”

(7) line 314 — is it not $t^* > 750$, not $t^* < 750$ as written??

Our response: The inequality sign was incorrect. It has been corrected. We thank the reviewer for spotting these smaller points.